



# Reviews and syntheses: Gaining insights into evapotranspiration partitioning with novel isotopic monitoring methods

Youri Rothfuss[1]★, Maria Quade[1]★, Nicolas Brüggemann[1], Alexander Graf[1], Harry Vereecken[1], and Maren Dubbert[2,3]

[1]Forschungszentrum Jülich GmbH, Institute of Bio- and Geosciences, Agrosphere Institute (IBG-3), Jülich, 52425, Germany
[2]Albert-Ludwigs-Universität Freiburg, Institut für Forstwissenschaften, Fakultät für Umwelt und natürliche Ressourcen, Freiburg, 79110, Germany
[3]Leibniz-Institut für Gewässerökologie und Binnenfischerei Berlin, Landscape Ecohydrology, Berlin, 12587, Germany
★These authors contributed equally to this work

**Correspondence**: Youri Rothfuss (y.rothfuss@fz-juelich.de)

**Abstract.** Disentangling ecosystem evapotranspiration (ET) into evaporation (E) and transpiration (T) is of high relevance for a wide range of applications, from land surface modelling to policy making. Identifying and analysing the determinants of the ratio of T to ET (T/ET) for various land covers and uses, especially in view of climate change with increased frequency of extreme events (e.g., heatwaves and floods), is prerequisite for forecasting the hydroclimate of the future and tackling present issues, such as agricultural and irrigation practices.

A powerful partitioning method consists in determining the water stable isotopic compositions of ET, E, and T ($\delta_{ET}$, $\delta_E$, and $\delta_T$, respectively) from the water retrieved from the atmosphere, the soil, and the plant vascular tissues. The present work emphasises the challenges this particular method faces (e.g., the spatial and temporal representativeness of the T/ET estimates, the limitations of the models used and the sensitivities to their driving parameters) and the progress that needs to be made in light of the recent methodological developments. As our review is intended for a broader audience beyond the isotopic ecohydrological and micrometeorological communities, it also attempts to provide a thorough review of the ensemble of techniques used for determining $\delta_{ET}$, $\delta_E$, and $\delta_T$, and solving the partitioning equation for T/ET.

From the current state of research, we conclude that the most promising way forward to ET partitioning and capturing the sub-daily dynamics of T/ET is in making use of non-destructive online monitoring techniques of the stable isotopic composition of soil and xylem water. Effort should continue towards the application of the eddy covariance technique for high-frequency determination of $\delta_{ET}$ at the field scale as well as the concomitant determination of $\delta_{ET}$, $\delta_E$, and $\delta_T$ at high vertical resolution with field-deployable lift systems.



## 1 Introduction

Only one fifth of the global hydrological flux occurs over the continents (Chahine, 1992; Dirmeyer et al., 2006). The water
cycle is therefore, from a purely quantitative point of view, predominantly an oceanic one. Yet it greatly shapes land surfaces
and controls their ecological functioning (Marotzke et al., 2017). Small relative changes in local precipitation regimes, as
induced by global warming, have tremendous repercussion on, for instance, summer climate variability (Seneviratne et al.,
2006) and people's local access to fresh water (Oki and Kanae, 2006; Milly et al., 2005).

Numerous studies also quantify the coupling strength between the land and the atmosphere: soil moisture content as well as
the evapotranspiration (ET) flux positively or negatively affect precipitation and surface temperature (Koster et al., 2004; Wei
and Dirmeyer, 2019; Zeng et al., 2017). They highlight the decisive role of the critical zone in regulating climate both locally
and globally (Grant and Dietrich, 2017).

A pivotal parameter in landscape hydrology and ecology is the transpiration (T) to evapotranspiration (ET) ratio (T/ET) (see
the reviews of Kool et al., 2014; Anderson et al., 2017; Stoy et al., 2019). Isolating the T flux in ET is of utmost importance
for a wide range of applications, because of its link to plant water uptake, for e.g., optimizing irrigation practices (Skaggs et
al., 2010), tackling ecological questions in water-limited ecosystems (Rothfuss and Javaux, 2017), or for a better representation
of the relations between the carbon and water cycles in climate models (Humphrey et al., 2018; Ito and Inatomi, 2012). At the
global scale, the uncertainty of the T/ET estimate remains high; it has been estimated to range from 13-90 %, depending on
the source and type of data (e.g., satellite- or isotopic-based) and method (modelling or data reanalysis) (Lawrence et al., 2007;
Alton et al., 2009; Jasechko et al., 2013; Wang et al., 2014; Wei et al., 2017). Ultimately, this conditions the ability of land-
surface models to provide sensitivity of the overall ET flux to changes in precipitation and land cover (Wang and Dickinson,
2012).

Spatial and temporal variability add even more uncertainty to our knowledge on T/ET at the local scale, which is a prerequisite
for a meaningful use of such estimates for any of the practical and scientific questions mentioned above. Partitioning ET into
the raw components E and T at the field and sub-daily spatiotemporal scales is generally performed by an ensemble of
partitioning methods, which can be divided into instrumental approaches and correlation-based modelling approaches (Scanlon
and Kustas, 2010). The former approach includes, e.g., the eddy covariance (EC) technique (Baldocchi, 2014; Reichstein et
al., 2005), soil-flux chamber measurements (Raz-Yaseef et al., 2010; Lu et al., 2017), micro-lysimeter measurements (Kelliher
et al., 1992), or atmospheric profile measurements (Ney and Graf, 2018).

Another powerful instrumental method to partition ET is based on the analysis of its hydrogen or oxygen isotopic composition,
i.e., the water vapour atom ratio in rare ($^2$H or $^{18}$O) and abundant ($^1$H or $^{16}$O) stable isotopes and expressed on the international
"delta" ($\delta$) scale (Dubbert and Werner, 2019). The method utilizes the natural discrepancies in isotopic composition of the
ecosystem evaporation ($\delta_E$) and transpiration ($\delta_T$) fluxes. The difference $\delta_T - \delta_E$ originates primarily from thermodynamic and
kinetic fractionation during phase change and transport processes undergone by water evaporating from soil on the one hand,
and water extracted by a root system and transpired by the canopy on the other hand. The observed discrimination against





stable isotopologues along the soil-plant-atmosphere water path can be conceptualized two-fold, i.e., phase change- and diffusion-driven, and quantified by the so-called equilibrium and kinetic fractionations, respectively, for which we will later review the physically-based expressions. The term $(\delta_T - \delta_E)$ is also determined by

(i)      the difference in lower boundary conditions acting on E and T, i.e., the $\delta$-value of soil water at the evaporating front (EF) and leaf water at the transpiration site, respectively, and

(ii)      the prevalence (or non-prevalence) of isotopic steady state (ISS) for transpiration, i.e. whether $\delta_T$ is independent of time (Farquhar and Cernusak, 2005; Dubbert et al., 2014a). Note that the ISS assumption is generally not made for evaporation flux (but see for an exception: Rothfuss et al., 2010).

The T/ET fraction is obtained by inverting the isotopic mass balance equation $\delta_{ET} = (1 - T/ET)\delta_E + (T/ET)\delta_T$ (Yakir and Sternberg, 2000):

$$T/ET = \frac{\delta_{ET} - \delta_E}{\delta_T - \delta_E} \qquad (1)$$

Equation (1) highlights how the isotopic partitioning methodology differs from other instrumental approaches, such as based on a combination of different techniques (e.g., lysimeter and EC measurements): it solely relies on measurements and/or analytical modelling of the stable isotopic compositions of the components ET, E, and T. Behind this apparent simplicity, the isotopic partitioning methodology is limited in its application in different ways, such as the inability – until recently – to provide continuous (i.e., non-destructive) $\delta_E$, $\delta_T$, and $\delta_{ET}$ assessments. Part of these limitations were overcome with the availability of field-deployable laser-based spectrometers. These instruments allow for long-term monitoring of soil water vapour and plant transpiration isotopic compositions when combined with gas-permeable membrane or tubing technology (Beyer et al., 2020).

A variety of different methods exists to measure or estimate $\delta_E$, $\delta_T$ and $\delta_{ET}$. The central aim of this study is to identify the challenges the ensemble of isotopic methods currently face and how they should progress in the future (section 3). Particularly, the abovementioned emerging monitoring methods are reviewed for the specific purpose of ET partitioning. As such, our work differs from those of Wang and Yakir (2000), Yakir and Sternberg (2000), Xiao et al. (2018), and Sun et al. (2019). In addition and for non-specialists readers, we thoroughly review the underlying concepts and techniques involved in the determination of $\delta_E$, $\delta_T$ and $\delta_{ET}$. In order to highlight the important progresses made over the past 30 years, we also give a literature overview (section 2). Finally, section 5 presents a summary as well as the possible ways forward for the isotopic partitioning community.

## 2 Literature overview

A total of 39 studies were found by entering the search terms "(("evapotranspiration" or "transpiration" or "evaporation") and partition* and isotop*)" into the ISI Web of Science search engine (http://www.webofknowledge.com/). The reader will find a graphical summary in Fig. 1 as well as a detailed description for each of the entries in Table A2 of Appendix A. On average,



approximately 1.3 (2.4) partitioning studies were published each year over the period 1989-2007 (2008-2020) with an average annual citation rate of 12 (143) (Fig. 1a).

To the authors' knowledge, the first scientific article reporting on the possibility to partition ET on basis of the differences in isotopic composition of ecosystem ET, soil evaporation, and plant transpiration was that of Bariac et al. (1987). An attempt to

use this possibility was made in the study of Walker and Brunel (1990) (Table A2) –, of which ET partitioning was the not the major focus – but remained, according to the authors, not conclusive. Ten years later, Jean-Pierre Brunel and his colleagues could provide the first water stable isotope-derived estimation of the relative importance to ET of the transpiration of the tropical and water-stress resistant plant *Guiera senegalensis* (Brunel et al., 1997), which was noticeably low (approx. 20%). In the meantime, Moreira et al. (1997) applied the so-called 'Keeling plot' technique (Keeling, 1958) (see section 3.1) for

determination of the isotopic composition of ET for the specific purpose of partitioning. The isotopic compositions of soil E and plant T at two sites (one pasture, one forest) in the Amazon basin were inferred by using the atmospheric part of the Craig and Gordon (1965) model (see section 3.2) and by assuming steady state transpiration flux (see section 3.3), respectively. The authors could provide evidence of the strong prevalence of T in the ET budget. In a hybrid work coupling a review of the state of the art with field measurements, Wang and Yakir (2000) computed an exceptionally high T/ET ratio value (96.5-98.5%).

Hsieh et al. (1998) and Ferretti et al. (2003) opted for a common water isotope mass-balance equation applied at the plot scale, which they solved by making a series of simplifying hypotheses: atmospheric water vapour is in thermodynamic equilibrium with soil water, and the isotopic composition of T is the amount-weighted average of the isotopic compositions of precipitation and soil water. Ferretti et al. (2003) obtained T/ET values ranging between 10 and 60%, depending on the growing season, in a semi-arid grass steppe, while Hsieh et al. (1998) estimated T/ET to span from 14 to 71% as annual rainfall increased along

two sampling transects in Hawaii.

Yepez et al. (2003) applied the Keeling plot technique specifically to two distinct ecosystem layers of a savanna woodland in southern Arizona, USA, i.e., the understorey dominated by the *Sporobolus wrightii* $C_4$-grass and the canopy populated by the mesquite tree *Prosopis velutina*. By doing this, they could capture the isotopic composition of ET representative of each of the two ecosystem layers. In order to partition ET, the authors computed the isotopic composition of the whole ecosystem T as a

composite function of the isotopic compositions of grass and tree T fluxes. Finally, it was determined that grass and tree T amounted to 15 and 75 % of total ET. In a follow up study, Yepez et al. (2005) used large gas exchange chambers positioned either on bare soil plots or sparsely vegetated areas of a semi-arid grassland in Arizona, USA. They determined – again with the Keeling plot technique – the isotopic composition of E and ET following an irrigation pulse. This is, to the authors' knowledge, the first use of a closed chamber system in the context of ET partitioning, where T is the single source of the

change in air moisture concentration. In contrast to the previous partitioning studies, Yepez et al. (2005) determined the isotopic composition of the non-steady state (NSS) T flux on the basis of plant physiological and micro-meteorological measurements using the formulation of Farquhar and Cernusak (2005) (see also for later examples: Sun et al., 2014; Hu et al., 2014). Yepez et al. (2005) finally calculated T/ET values ranging between 35 and 43% the first three days after irrigation, and decreased to 22 % after one week. This showed the relative prevalence of E at the grassland site. In another study in semi-arid environmental



setting (Marrakech, Morocco), Williams et al. (2004) observed that irrigation enhanced soil E of an olive orchard (*Olea europaea* L.). Mid-day T/ET average decreased from approx. 100% (determined prior irrigation) to 69-86% (computed over the 5-day period after irrigation). Xu et al. (2008) investigated the discrepancies between T/ET assessments from either $\delta^2$H of $\delta^{18}$O data collected in a subalpine shrubland (Balang Mountain, China). As for Yepez et al. (2005), they calculated the uncertainties linked with determination of T/ET with the Isoerror software (Phillips and Gregg, 2001). Furthermore, they could
differentiate between canopy (*Quercus aquifolioides*) and understory (e.g., *Cystopteris montana*) contributions to ET by using the multi-source mixing model Isosource (Phillips and Gregg, 2003).

Wenninger et al. (2010) and Sutanto et al. (2012) used similar semi-controlled experimental setups equipped with soil liquid water (rhizon) samplers. They simulated the contributions to ET of soil E, plant T and canopy interception using an isotopic mass-balance model. In their framework, the destructive sampling of soil to retrieve the isotopic composition of soil E was not
needed, while a number of simplified hypotheses had to be made regarding T. Wenninger et al. (2010) simulated a T/ET ratio value of 70% for teff (*Eragrostis tef*) during the course of their experiment. Sutanto et al. (2012) found a comparable value for a grass cover (T/ET = 87%). In both of these studies, the isotopic partitioning results were confronted with additional (e.g., micro-meteorological) measurements and independent models such as HYDRUS-1D. In two companion papers (Rothfuss et al., 2010; 2012), T/ET of a 0.2 m² surface area monolith was computed and simulated for five selected dates under strictly
controlled conditions in a climate chamber along the development of a tall fescue cover (*Festuca arundinacea*). T/ET was determined to increase from 6% (16 days after sowing) to 95% (43 days after sowing). Rothfuss et al. (2012) further confronted the isotopic data with simulations with the SiSPAT-Isotope model (Braud et al., 2005). One year earlier, Haverd et al. (2011) used another isotopically enabled soil-vegetation-atmosphere transfer (SVAT) model, Soil-Litter-Iso (Haverd and Cuntz, 2010), using data from a field experiment (Eucalyptus forest, south eastern Australia) in a similar framework, i.e., by running
a multi-objective calibration to estimate a given set of model parameters. However, in contrast to Rothfuss et al. (2012), they could show that the added information provided by the isotopic data ($\delta^2$H) was not effective in better constraining the model for determination of T/ET (in their case equal to 85 ±2%). Another simulation study was published by Pei Wang et al. (2015), where a physically-based model solving the energy and water balance in the soil-plant-atmosphere continuum (Wang and Yamanaka, 2014) was coupled to an isotopic module accounting for fractionation processes during E and T. Noticeably, Wang
et al. (2015) provided the first test of the 'Peclet effect' (Farquhar and Lloyd, 1993), formalizing in a physically-based matter the compartmentalization of leaf water (see also Piayda et al. (2017) for another example). Wang et al. (2015) simulated T/ET of a temperate grassland to spread over a wide range of values (i.e., 2-99%) during the course of a 190 day-long experiment. Wei et al. (2018) used a similar modelling framework as in Wang et al. (2015) and found that the 3-months ET-weighted T/ET values were equal to 74, 93, and 81 % for three different crops, i.e. rice, corn and wheat, respectively, grown in temperate
(rice, Japan) and semi-arid monsoonal (corn and wheat, China) environmental conditions.

Wang et al. (2010; 2013) introduced the use of closed chambers to determine by mass balance the isotopic compositions of ecosystem water fluxes (E, T, and ET) in a non-destructive way (Wang et al., 2010; 2013). This allowed the authors not to rely on either (i) making the assumption of T at ISS for partitioning ET fluxes or (ii) modelling the isotopic composition of T at





NSS (see chapter 3.3). They also published the first ET partitioning study where water vapour hydrogen and oxygen isotopic

compositions were measured online with an infrared laser spectrometer. Wang et al. (2010) calculated T/ET values for the mesquite tree (*Prosopis chilensis*) grown under controlled conditions (Biosphere 2 facililty, Arizona, USA, see for details: Barron-Gafford et al., 2007 ), ranging from 61 to 83% at 25 and 100% woody cover, respectively. Wang et al. (2013) compared T/ET ratio (65-77% vs. 83-86%) computed from control vs. warming plots, taking advantage of a long-term grassland multiple-factor climate control experiment in Oklahoma, USA.

Bijoor et al. (2011) investigated the partitioning of ET in a freshwater marsh dominated by *Typha latifolia* in California, USA. They found a good agreement between T/ET values estimated on the one hand from isotopic analysis and from micro-meteorological (e.g., EC) measurements on the other, while they highlighted the high uncertainty of the isotope estimates (i.e., standard error value > 37%). Zhang et al. (2018) investigated another marsh wetland in China and found out that the two dominant species (*Scirpus triqueter* and the invasive *Phragmites australis*) contributed each equally (20 %) to ET flux.

Dubbert et al. (2013) quantified the sensitivity of the partitioning of ET to a number of factors (e.g., value of the kinetic fractionation factor, assumption of steady-state T) during a field experiment in central Portugal. They also compared direct measurements of the isotopic composition of E (with gas exchange chambers coupled to a laser spectrometer) to simulations with the Craig and Gordon (1965) model. Similar to Rothfuss et al. (2010; 2012), the authors underlined the need to complement isotopic measurement with micro-meteorological and physiological observations. In the same open cork-oak

(*Quercus suber* L.) savanna, Dubbert et al. (2014b) investigated the impact of the understorey vegetation (annual grass and forbs) on the total ecosystem water budget. They could discriminate between T of trees and grass and highlighted the stability of the former throughout the year and the strong decrease of the latter during the summer. Piayda et al. (2017) differentiated between open and shaded portions of the same experimental site and found T/ET ranging between 9 to 59% and between 17 to 66%, respectively. Good et al. (2014) studied the uncertainty of the T/ET values obtained for a grassland site, following a

30 mm isotopically enriched irrigation event, e.g., as a function of the uncertainty linked with the estimate of $\delta_{ET}$ obtained with the Keeling plot technique (according to Good et al., 2012). The authors found on average a value of 30 (±5) % for T/ET over the 15 days of the experiment.

Hu et al. (2014) determined a mean T/ET value of 83% in a semi-arid shrubland in China dominated by *Stipa kryroii* and *Artemisia frigida*. They tested for the first time the so-called flux-gradient approach (Lee et al., 2007; see section 3.1) for

determination of $\delta_{ET}$. They argued that the uncertainty of the $\delta_{ET}$ estimates had the strongest effect on T/ET uncertainty(?) in their case. Also Wei et al. (2015) found that the greatest source of uncertainty of T/ET of a rice paddy field was linked to the determination of $\delta_{ET}$, this time using the Keeling plot technique. They could describe T/ET as an exponential function of leaf area index (LAI) (i.e., T/ET[%]=67·LAI$^{0.25}$). Wu et al. (2017) found slightly different parameters of the same LAI model (71·LAI$^{0.14}$) for a maize crop grown under semi-arid conditions (Gansu Province, China).

Berkelhammer et al. (2016) compared the outcome of the isotopic partitioning method with EC-derived T/ET values. They underlined the goodness of fit of the two methods as well as the stability of the T/ET ratio as a function of LAI over multiannual time scales. Wen et al. (2016) investigated the contribution of spring maize T to ET in an arid artificial oasis part of the Heihe





river catchment (China) and reported it to be quite constant (mean T/ET value of 87 ± 5.2 %). Collected data was further used by Zhou et al. (2018) and Xiong et al. (2019). Zhou et al. (2018) showed similarities between results of the isotopic partitioning

method and a coupled approach of EC and lysimeter data. They underlined, however, that both methods simulate higher T/ET ratios, with poor temporal dynamics not reflecting those of leaf area index, than their benchmark method, i.e., based on the incorporation of vapour pressure deficit into the expression of the water-use efficiency concept. Xiong et al. (2019) observed a good match between T/ET daily values (54-97%, with a mean value of 85 %) as obtained with their isotope method and with a net radiation and temperature-dependant model coupled to imaging radiometry.

Aouade et al. (2016) found a decreasing diurnal (i.e., morning vs. afternoon) amplitude of T/ET in a winter wheat field in Morocco under wet conditions after flood irrigation with a soil water content of approx. 0.35 m$^3$ m$^{-3}$, and the opposite under dry conditions with a soil water content value of 0.15 m$^3$ m$^{-3}$. Aouade et al. (2020) compared the T/ET results for dry conditions of Aouade et al. (2016) to independent assessments using the Interaction between Soil, Biosphere, and Atmosphere (ISBA) model (Masson et al., 2013) and found that they were within the same range (73-89 %). The study of Lu et al. (2017) focused

on the efficiency of irrigation strategies in southern California (USA). They document that the investigated field of *Sorghum bicolor* was responsible for 46% of water consumption following the irrigation event. Quade et al. (2019) applied for the first time a non-destructive approach for determination of the isotopic composition of E by sampling the soil water vapour with gas-permeable tubing and measuring its isotopic composition online with a laser spectrometer as described in Rothfuss et al. (2013). They cross-compared the T/ET values obtained with different methods including the combination of EC and lysimeter

flux data, and isotopic assessments based on either water $\delta^2$H or $\delta^{18}$O data. They further compared T/ET results obtained on the basis of the non-destructive method of Rothfuss et al. (2013) (for determination of the isotopic composition E) with those of traditional destructive soil sampling in combination with the Craig and Gordon (1965) model. They found significant differences in T/ET of a sugar beet (*Beta vulgaris*) field between the different methods on four days at different stages of the canopy development (0.7 < LAI < 6.7).

In a review on the use of water stable isotope analysis for determination of plant root water uptake dynamics (Rothfuss and Javaux, 2017), the authors underlined the need for field studies in croplands. This is not the conclusion of the present literature overview, as the three main land surface types, i.e., cropland, forests, and grassland (in monoculture or mixed culture) are rather equally represented with a relative proportion of 33, 32, and 41%, respectively (Fig. 2b). More than one third of the scientific publications analysed in the present review (i.e. 38 %) applied the isotopic methodology in semi-arid or desert regions

(Fig. 1c). Nevertheless, a wide range of climate types (e.g., subtropical-humid, Mediterranean or subarctic, Fig. 1c) as well as regions (e.g., Northern America, sub-Saharan Africa or Eastern Asia, Fig. 1d) is investigated as well. 30 of the 39 reviewed studies were conducted in the field, and only eight (21 %) used a physically-based numerical model to simulate T/ET ratios on the basis of the collected isotopic (and water status) data (Fig 1e). Furthermore, 95 % of the field studies were conducted at natural isotopic abundance, either under normal precipitation regime (85 %) or in the framework of an irrigation experiment

(10 %). The remaining 5 % of studies (Yepez et al., 2005; Good et al., 2014) applied a labelling pulse of $^2$H-enriched water to the soil for better discrimination between the three terms of the mixing equation (Eq. (1)).



There is naturally a strong link between the temporal resolution in T/ET estimates and the temporal extent of the T/ET time series (Fig. 1b). The vast majority of the studies (85 %) provided T/ET values at hourly to subweekly resolution over periods of time not exceeding a few months. This is partly a sign of the limitation of the isotopic methodology, which was mentioned

in the introduction, i.e., the labour-intensive and time-consuming destructive sampling of soil and plant material and the subsequent water extraction step. In two studies only (Hsieh et al., 1998; Ferretti et al. 2003), authors could calculate T/ET at weekly to monthly resolution overs several years. For doing this, they made a series of abovementioned simplifying hypotheses, which allowed them, amongst other things, not to rely on sampling of plant material, thereby significantly saving extraction and analysis time. The authors of the present work note that, on the other hand, the question of spatial variability or

representativeness of the T/ET estimates is rarely addressed in the literature (but see section 3.1 for the issue of spatial representativeness of $\delta_{ET}$).

## 3 Methodological review

In this section, the methods used for determination of the three terms in the partitioning equation (Eq. 1), i.e., $\delta_{ET}$, $\delta_E$ and $\delta_T$ for final computation of the T/ET ratio will be covered (subsections 3.1.1, 3.2.1, and 3.3.1, respectively), with special emphasis

on challenges and new technical and methodological developments (subsections 3.1.2, 3.2.2, and 3.3.2, respectively). Three main approaches emerge from the analysis: $\delta_{ET}$, $\delta_E$ and $\delta_T$ can be either determined by

(i)      solving the mass balances for the different water vapour isotopologues,

(ii)     using physical models based on macroscopic analogies of Ohm's law, or

(iii)    using a statistical framework (Fig. 2).

Note that it is not the present work's intention to give a thorough review of the physically-based and isotope-enabled soil-vegetation-atmosphere numerical models used by Haverd et al. (2011), Sutanto et al. (2012), and Rothfuss et al. (2012) for simulation of T/ET. For this, the readers may refer also to Haverd and Cuntz (2010) and Braud et al. (2005). Likewise, the authors choose not to describe one particular ensemble of methods in detail (used in seven different studies, see Table 2 and referred to as "water balance" in Fig. 1.) based on solving a water mass-balance equation and not relying on the sampling or

monitoring of plant and soil water, and atmospheric water vapour.

### 3.1 Isotopic composition of evapotranspiration

### 3.1.1 Methods

The prevalent method (43 % of the reviewed studies, Fig. 1h) for determining the isotopic composition of ET is based on solving a mass balance equation (Fig. 2a-c). It was named after Charles D. Keeling who originally used it to quantify the $CO_2$

carbon isotopic composition in the atmosphere as a linear function of the reciprocal of the $CO_2$ concentration (Keeling, 1958). The so-called 'Keeling plot' technique simply considers that the water vapour measured in some ecosystem atmosphere



compartment (of concentration $C_{atm}$, dimension of M L$^{-3}$) originates from two sources, namely (i) the background water vapour (of concentration $C_{bg}$ [M L$^{-3}$]), transported advectively, and (ii) evapotranspiration ET (of concentration $C_{ET}$ [M L$^{-3}$]):

$$C_{atm} = C_{bg} + C_{ET}. \tag{2}$$

Practically, laser-based spectrometers measure water vapour volume mixing ratio, $\chi$[-], the ratio of water vapour pressure and total (dry) atmospheric pressure:

$$\chi_{atm} = \chi_{bg} + \chi_{ET}. \tag{3}$$

A similar equation can be written for stable isotopes:

$$\delta_{atm}\chi_{atm} = \delta_{bg}\chi_{bg} + \delta_{ET}\chi_{ET}, \tag{4}$$

with $\delta_{atm}$ and $\delta_{bg}$ the isotopic compositions of the ambient air and background air, respectively. Combining Eqs. (3) and (4) and rearranging for $\delta_{atm}$ leads to the following expression (Eq. (5), see Fig. 2 for an illustration):

$$\delta_{atm} = \frac{1}{\chi_{atm}}\left[\chi_{bg}(\delta_{bg} - \delta_{ET})\right] + \delta_{ET}. \tag{5}$$

To the conditions that

(i)       both $\chi$ and $\delta$-values of the background water vapour and ET remain constant during the measurement period and

(ii)       there is no loss of water vapour from the atmosphere (e.g., during dewfall),

it is possible to determine $\delta_{ET}$ as the Y-intercept of the regression line of the relationship between $\delta_{atm}$ and $1/\chi_{atm}$. In this framework the sign of the linear regression slope $s$ [M L$^{-3}$] $= C_{bg}(\delta_{bg} - \delta_{ET})$ is therefore constrained by the difference $(\delta_{bg} - \delta_{ET})$; $s$ is generally negative, apart from some bare soil situations (Yakir and Sternberg, 2000) (Fig. 3a). Note that it is also possible to derive $\delta_{ET}$ by inverting the expression for $s$, although, to our knowledge, such a possibility has not yet been tested

in the literature, certainly because the determination of $C_{bg}$ and $\delta_{bg}$ is not straightforward in the field.

One important prerequisite for the applicability of the Keeling plot is a significant span in $\chi_{atm}$ values over the course of the measurements (Fig. 3b-c). High $\chi_{atm}$ values are especially needed to reduce the statistical uncertainty of $\delta_{ET}$ (Good et al., 2012). In case of a single observation height (Wei et al., 2018; Good et al., 2014; Wei et al., 2015), the time factor is critical. $\chi_{atm}$ variations should not be obtained at the expense of the validity of the aforementioned core assumption (i), i.e., steady values

of ET and background $\chi$ and $\delta$. Another option beside the one just described, which we could refer to as the 'temporal Keeling plot' technique, is to drastically increase the span of $\chi_{atm}$ values by collecting data at different observation heights during a short period of time (~ 1 hour), which could be referred to as 'spatial Keeling plot'. From our literature compilation, the spatial Keeling plot is preferred over the temporal one (i.e., 32 vs. 7 studies).

Another technique (18 % of the reviewed studies, Fig. 1h) for determining $\delta_{ET}$ requires the manipulation of transparent chamber

systems to enclose and tightly seal the soil and vegetation (e.g., Yepez et al., 2005; Piayda et al., 2017). Two different applications exist, both based on the mass balance approach. In the first one (referred to as "Chamber (InOut)" in Table A2



and Fig. 1h), the chamber is flushed with ambient air, and $\delta_{ET}$ is deduced from the difference in water vapour mixing ratio and isotopic composition measured alternatingly at the inlet (subscript 'in') and outlet (subscript 'out') of the chamber (e.g., Wang et al., 2013; Dubbert et al., 2013):

$\qquad \delta_{\text{out}}\chi_{\text{out}} = \delta_{\text{in}}\chi_{\text{in}} + \delta_{\text{ET}}\chi_{\text{ET}}.$ $\hfill (6)$

Equation (6) is strictly valid only for conservative flow conditions. In other studies (e.g., Dubbert et al., 2014b), the change in flow rate ($u$ [$L^3 T^{-1}$]) between in- and outlet due to the addition of water vapour originating from the soil and/or the plant is taken into account as follows:

$\qquad \delta_{\text{out}}\chi_{\text{out}}u_{\text{out}} = \delta_{\text{in}}\chi_{\text{in}}u_{\text{in}} + \delta_{\text{ET}}\chi_{ET}(u_{\text{out}} - u_{\text{in}}).$ $\hfill (6')$

By conservation of dry air flow, i.e. $u_{\text{out}}(1 - \chi_{\text{out}}) = u_{\text{in}}(1 - \chi_{\text{in}})$ (Simonin et al., 2013), Eq. (6') becomes

$\qquad \delta_{\text{ET}} = \frac{\chi_{\text{out}}\delta_{\text{out}} - \chi_{\text{in}}\delta_{\text{in}}}{\chi_{\text{out}} - \chi_{\text{in}}} - \frac{\chi_{\text{in}}\chi_{\text{out}}(\delta_{\text{out}} - \delta_{\text{in}})}{\chi_{\text{out}} - \chi_{\text{in}}}.$ $\hfill (7)$

The second term on the right-hand side of Eq. (7) therefore accounts for the increase of flow rate due to ET in the chamber. An alternative consists in flushing the chamber with dry air instead of ambient air, so that the isotopic composition of the outlet water vapour directly reflects that of ET. In the second application (named "Chamber (Keeling Plot)" in Fig. 1h), the chamber

is flushed in a closed loop with ambient air, and $\delta_{ET}$ is obtained by linear regression of the isotopic composition of the chamber air versus the inverse of the water vapour mixing ratio using the Keeling (1958) plot technique.

In 10% of the referenced studies (Wen et al., 2016; Wei et al., 2018; Zhou et al., 2018), authors determined $\delta_{ET}$ values by analogy to Ohm's law. The so-called 'flux gradient method' (Lee et al., 2007) is based on the premise that the ET flux density rate ($F_{ET}$ [$L^3 L^{-2} T^{-1}$, expressed typically in mol m$^{-2}$ s$^{-1}$]) is proportional to $\Delta\chi_{\text{atm}}/\Delta z_{\text{atm}}$ [$L^{-1}$, typically in m$^{-1}$], the gradient of

water vapour mixing ratio between two observation heights (with $z_{\text{atm}}$ standing for height):

$\qquad F_{\text{ET}} = -K \frac{\rho_{\text{atm}}}{M_{\text{atm}}} \frac{\Delta\chi_{\text{atm}}}{\Delta z_{\text{atm}}}.$ $\hfill (8)$

The water vapour transport is determined by the overall conductance of the air boundary layer expressed here as $-K \rho_{\text{atm}}/M_{\text{atm}}$ with $\rho_{\text{atm}}$ [$M L^{-3}$] and $M_{\text{atm}}$ [$M L^{-3}$, units of kg mol$^{-1}$] the dry air volumetric mass and molecular weight, and $K$ [$L^2 T^{-1}$] the eddy diffusivity of water vapour. The isotopic ratio of ET ($R_{ET}$ [–]), which can be defined as the ratio of the flux density rates of the

rare (superscript $i$) and abundant (superscript $j$) isotopologues ($^iF_{ET}$ and $^jF_{ET}$, respectively), can be therefore expressed as

$\qquad R_{\text{ET}} = {}^iF_{\text{ET}}/{}^jF_{\text{ET}} \approx \Delta^i\chi_{\text{atm}}/\Delta^j\chi_{\text{atm}},$ $\hfill (9)$

assuming that differences in $K$ among water stable isotopologues are not significant, i.e. $^iK = {}^jK = K$ (Yakir and Wang, 1996; Griffis et al., 2005). $^i\chi_{\text{atm}}$ and $^j\chi_{\text{atm}}$ are the water vapour mixing ratio of rare and abundant isotopologues, respectively. Equation (9) can be further rearranged as

$\qquad {}^i\chi_{\text{atm}} = R_{\text{ET}}{}^j\chi_{\text{atm}} + C,$ $\hfill (10)$





where $R_{ET}$ is the slope and $C$ [-] the y-intercept of the linear relationship between ${}^i\chi_{atm}$ and ${}^j\chi_{atm}$. Equation (10) becomes in $\delta$-notation:

$$\delta_{atm} = \delta_{ET} + C/R_{std}\frac{1}{{}^j\chi_{atm}} \tag{11}$$

by dividing its left and right terms by ${}^j\chi_{atm}R_{std}$ with $R_{std}$, the isotopic ratio of the internationally accepted water standard,

namely the Vienna Standard Mean Ocean Water (V-SMOW) (Gonfiantini, 1978). We note that, by assuming ${}^j\chi_{atm} \approx \chi_{atm}$, the flux gradient and Keeling plot techniques are mathematically identical with $C = \chi_{bg}(\delta_{bg} - \delta_{ET})R_{std}$.

Griffis et al. (2010) and Good et al. (2012) used a combination of the EC technique and infrared tunable diode laser (TDL) water isotope spectroscopy to derive $\delta_{ET}$ values from simultaneous changes in wind velocity ($\omega$ [L T$^{-1}$]) and ${}^i\chi_{atm}$. In this statistical framework and by

(i)       considering that air density and storage fluctuations are negligible during the measurement period (typically 30 minutes) and

(ii)      changing the coordinate system so that the vertical wind velocity mean value ($\overline{\omega}$) equals zero,

$F_{ET}$ is expressed as:

$$F_{ET} = \frac{\rho_{atm}}{M_{atm}}\overline{\omega'\chi_{atm}'}. \tag{12}$$

The term $\overline{\omega'\chi_{atm}'}$ is the average (overbar symbol) product of the differences between instantaneous and mean values (indicated by the prime symbols) of wind velocity and water vapour mixing ratio, in other words the covariance between the $\omega$ and $\chi_{atm}$ monitored variables. Similar to Eq. (11), we obtain after converting in $\delta$-notation the expression for the isotopic composition of ET:

$$\delta_{ET} = \frac{{}^iF_{ET}/{}^jF_{ET}}{R_{std}} - 1 = \frac{\overline{\omega'{}^i\chi_{atm}'}/\overline{\omega'{}^j\chi_{atm}'}}{R_{std}} - 1 \tag{13}$$

An alternative to Eq. (13) consists in considering the high-frequency variations of $\delta_{atm}$ rather than those of the individual mixing ratios ${}^i\chi_{atm}$ and ${}^j\chi_{atm}$. For this the isoflux (Lee et al., 2009), defined as $\overline{\omega'\delta_{atm}'}$ [L$^3$ L$^{-2}$ T$^{-1}$], is introduced:

$$\delta_{ET} = \frac{\overline{\chi_{atm}}}{\overline{\omega'\chi_{atm}'}}\overline{\omega'\delta_{atm}'} + \overline{\delta_{atm}} \tag{14}$$

### 3.1.2   Progress and challenges

Good et al. (2012) provided a comprehensive comparison of the various techniques (Keeling plot, flux gradient and EC) for

determination of $\delta_{ET}$. In addition to the temporal and spatial Keeling plot variations, they tested a third option where, instead of instantaneous measurements of $\delta_{atm}$ and $\chi_{atm}$ collected during 30 min, the mean values of $\delta_{atm}$ and $\chi_{atm}$ are calculated at each observation height (n=4) and used for regression. After a detailed uncertainty analysis, they concluded that the use of mean values instead of individual data points increased the uncertainty associated with $\delta_{ET}$, regardless of the kind (temporal vs.





spatial) of Keeling plot. In addition, the temporal and spatial Keeling plot techniques yielded significantly different values of

$\delta_{ET}$ for the same time interval. The authors could not conclude which value was the most representative. In addition, they found

a good agreement between the Keeling plot technique, applied at different heights, and the flux gradient method due to the

aforementioned mathematical similarities.

As previously mentioned, Yepez et al. (2005) and Wang et al. (2013) combined the Keeling plot technique with their closed

chamber systems. During the course of measurement (e.g., 6 min in Yepez et al. (2005)) and for the Keeling plot approach to

be valid, the increase of chamber water vapour mixing ratio (10-15 mmol mol$^{-1}$ in Yepez et al., 2005) should not induce

changes in both ET flow rate and isotopic composition. The fulfilment of this requirement of the Keeling plot technique is

verified in a first approach by the very existence of a linear relationship between chamber air $1/\chi$ and $\delta$-values.

Another issue related to the use of chamber systems is the occurrence of water vapour condensation on the inside of the

chamber or within the tubing system, e.g., following changes of incoming solar radiation during measurement. This may result

in eventual isotopic fractionation leading to unreliable (i.e., unstable and underestimated) observations of chamber air $\delta$-values.

To avoid such problems, the volume of the chamber is critical (i.e., the bigger the less sensitive to abrupt changes of outside

conditions) and active ventilation is mandatory. Ventilation not only prevents from condensation problems and pressure

anomalies (Longdoz et al., 2000) but also guarantees the prevalence of turbulent mixing conditions in the chamber. The latter

may not be ensured by a high turnover rate alone, i.e., the ratio of chamber volume and flow rate of flushed air. It is an

important prerequisite of the application of both techniques based on the Keeling plot and alternating measurements of the

water vapour mixing ratio and isotopic composition of inlet and outlet air (Eq. 7).

Measurements with dynamically purged chambers, which are combined with the latter type of mass balance applications, may

reduce the problem of condensation inside the chamber, which are combined with the latter type of mass balance applications,

may reduce the problem of condensation inside the chamber. A possibility is to flush the chamber with dry air, so that the

increase in water vapour mixing ratio and (positive or negative) change of isotopic composition measured at the outlet relative

to the inlet directly reflect the volume and isotopic composition of the moisture added by ET. Stable measurements over a

certain time period, depending on both chamber volume and inflow rate, would indicate ISS, and $\delta_{ET}$ may be directly measured

without any further calculations (e.g., Wang et al., 2010). However, dry air can stress the enclosed plants by artificially

increasing the chamber air vapour pressure deficit, which ultimately can result in NSS conditions. In this case, a steady increase

of chamber air $\chi$ should not be observed during the course of the measurement as it would be a sign of a significant difference

of micrometeorological conditions (temperature, vapour pressure deficit, and wind speed values) between ambient and

chamber air.

Both the temporal Keeling plot and the flux-gradient techniques suffer from the need of a high spatial gradient in the water

vapour mixing ratio and isotopic composition between the soil/canopy surface and the free atmosphere to obtain precise values

of $\delta_{ET}$. One possible way forward is to use a small (~few meters high) field lift system, the modus operandi of which is based

on the principles established by Mayer et al. (2009) and Noone et al. (2013), for a continuous monitoring of atmospheric height

profiles of the water vapour isotopic composition. To the authors' knowledge, only one study on an evergreen forest made use





of the principle in the context of ET partitioning (Berkelhammer et al., 2016). Ney and Graf (2018) designed a portable lift system for measuring the atmospheric water vapour and $CO_2$ mixing ratios in the field for various crops at a half-hourly

temporal resolution. Their system should allow for measuring highly vertically resolved water vapour isotopic profiles. For this, however, high-throughput and high-frequency isotopic analysers are needed to provide reliable information on ecosystem fluxes. Commercially available cavity ring-down laser spectrometers operate at low flow rate ($\varphi$) and frequency ($f$) (e.g., $25 \leq \varphi \leq 35$ ml min$^{-1}$ and $f \approx 1$ Hz for the L2120-$i$, L2130-$i$ and L2140-$I$ by Picarro, Inc., Santa Clara, CA, USA) and are, thus, not suitable for such measurements. To our knowledge only two instruments are able to monitor water vapour stable isotopic

compositions at higher flow rate and higher frequency: the lead-salt tunable diode laser spectrometer TGA200A (Campbell Scientific, Inc., Logan, Utah, USA; $\varphi = 1.7$ l min$^{-1}$ at $f = 10$ Hz) and the Quantum Cascade Laser (QCL) Trace Gas Monitor (Aerodyne, Inc., Billerica, MA, USA; $\varphi \leq 250$ l min$^{-1}$ and $f = 10$ Hz), although for the latter no published results are available. These instruments could be used for EC measurements of water vapour isotopologues, which are not common within the isotopic source-partitioning community. To the authors' knowledge, only two studies (Griffis et al., 2010; Griffis et al., 2011)

demonstrated that water vapour mixing ratio and ET measured with the traditional EC technique (with infrared laser analyser, e.g., LI-7500, Licor, Inc., Lincoln, NB, USA) agreed well with the combined EC/TGA200 measurements, which suggests that the measured ET isofluxes were realistic. The important advantage of the EC isotope technique resides in its ability to provide $\delta_{ET}$ estimates at the field scale and therefore demarks itself from the plot-scale Keeling plot and chamber-based approaches. Especially the spatial Keeling plot approach suffers from the fact that the different heights at which the isotopic composition

of water vapour is measured are representative for differently large areas of the studied ecosystem. The main disadvantage is that the aforementioned instruments are quite large and difficult to handle in situ. In addition, they need stable environmental conditions (especially temperature) during field deployment.



### 3.2 Isotopic composition of evaporation

#### 3.2.1 Methods

Two options are found in the literature (Fig. 1i) for determining the isotopic composition of the E flux, $\delta_E$:

(i)    by solving one of either mass balance equations (Eqs (7) or (11), see section 3.1) in combination with dynamically purged closed bare soil chambers (15 % of the reviewed studies, e.g., Dubbert et al., 2013; 2014b).

(ii)    by solving the so-called 'Craig and Gordon equation' (Eq. (18) below), which is derived from the atmospheric part of a transport model of water stable isotopologues, based on an analogy to Ohm's law (Craig and Gordon, 1965) (69

% of the studies), or

The two approaches differ in numerous aspects: while the first is non-destructive and requires on-line and continuous measurements of a few variables (i.e., water vapour mixing ratio and isotopic composition of the chamber inlet and outlet air), the second relies on destructive sampling of the soil and offline analysis of the extracted water. The Craig and Gordon equation demarks itself from Eqs. (7) and (11) also due to its complex parametrization. Craig and Gordon (1965) classically interpreted

the temporal changes in $\delta_E$ of a free water body with the help of a linear resistance model. We will shortly present the widely used model variation for water bound to the soil media (For an in-depth review, the reader is kindly referred to Horita et al., 2008). The only significant difference to the original model is the evaporating front vertical coordinate ($z_{EF}$), which may not correspond to that of the soil surface depending on the evaporation stage (Or et al., 2013; Merz et al., 2018). The isotopic ratio of evaporation, $R_E$, is expressed as the ratio of $^iF_E$ and $^jF_E$, i.e., the water vapour flux density rates [L$^3$ L$^{-2}$ T$^{-1}$] in rare and

abundant isotopologues, respectively, originating from the EF:

$$R_E = \frac{^jF_E}{^jF_E} = \frac{\Delta^i\chi_{atm}}{\Delta^j\chi_{atm}} = \frac{1}{^ir/^jr} \cdot \frac{^i\chi_{atm}(z_{EF}) - ^i\chi_{atm}(z_{atm})}{^j\chi_{atm}(z_{EF}) - ^j\chi_{atm}(z_{atm})}. \tag{15}$$

We note that Eq. (15) is analogous to Eq. (9) (Lee et al., 2007, see section 3.1), with the exception that the bulk resistances to vapour transport of the rare and abundant isotopologues ($^ir$ and $^jr$ [T L$^{-1}$], respectively) are not assumed equal. It follows from the fact that $^ir$ and $^jr$ relate to the air layer delimited between $z_{EF}$ and $z_{atm}$ (and not between the two observation heights in Eq.

(9)) where not only purely turbulent transport or eddy diffusivity, but also molecular diffusion and laminar flow are relevant. Furthermore, Craig and Gordon (1965) conceptualized the existence of a water vapour-saturated (superscript 'sat') air layer at the EF where isotopic thermodynamic equilibrium prevails:

$$^j\chi_{atm}(z_{EF}) = ^j\chi_{atm}^{sat} \tag{16a}$$

$$^i\chi_{atm}(z_{EF}) = ^i\chi_{atm}^{sat} = ^j\chi_{atm}^{sat}R_{sat} = ^j\chi_{atm}^{sat}\frac{R_{EF}}{\alpha_{eq}} \tag{16b}$$

where $R_{sat}$ and $R_{EF}$ are the isotopic ratios of the saturated air layer and of the soil liquid water at the EF, respectively. $\alpha_{eq}$ [-] is the isotopic equilibrium fractionation factor, first empirically determined by Majoube (1971) and later by Horita and Wesolowski (1994). $\alpha_{eq}$ depends on the soil temperature at the EF:



$$\alpha_{\text{eq}}(T_{\text{EF}}) = \exp\left(\frac{A}{T_{\text{EF}}^2} + \frac{B}{T_{\text{EF}}} + C\right) \tag{17}$$

with constants $A = 24844$, $B = -76.248$ and $C = 0.052612$ for $^2$H and $A = 1137$, $B = -0.4156$ and $C = -0.0020667$ for $^{18}$O. Craig
and Gordon (1965) identified the ratio of bulk resistances $^i r / {}^j r$ as the isotopic kinetic fractionation factor ($\alpha_K$ [-]). Finally,
by

(i)       considering that $^i\chi_{\text{atm}}(z_{\text{atm}}) = {}^j\chi_{\text{atm}}(z_{\text{atm}})R_{\text{atm}}$,

(ii)      dividing Eq. (15)'s right hand term numerator and denominator by $^j\chi_{\text{atm}}^{\text{sat}}$, and

(iii)     converting $R_E$ into $\delta_E$, we obtain:

$$\delta_E = \frac{1}{\alpha_K(1-h_{\text{atm}})}\left(\frac{\delta_{\text{EF}}+1}{\alpha_{\text{eq}}} - (\delta_{\text{atm}} + 1)h\right) - 1, \tag{18}$$

where $h_{\text{atm}}$ [-] is the relative humidity of the ambient atmosphere measured at vertical coordinate $z_{\text{atm}}$ and defined as the ratio
$^j\chi_{\text{atm}}(z_{\text{atm}})/{}^j\chi_{\text{atm}}^{\text{sat}}$. The possible difference in temperature measured at $z_{\text{atm}}$ and $z_{\text{EF}}$ should be accounted for by normalizing
$h_{\text{atm}}$ to the saturated vapour pressure ([M L$^{-1}$ T$^{-2}$], usually expressed in Pa) at the temperature of the EF (Rothfuss et al., 2010;
Quade et al., 2019).

Craig and Gordon (1965) argued that the kinetic fractionation factor was inversely proportional to the ratio of the molecular
diffusivities of $^1$H$_2^{16}$O ($^j D$) and of either $^1$H$^2$H$^{16}$O or $^1$H$_2^{18}$O ($^i D$):

$$\alpha_K = \frac{{}^i r}{{}^j r} = \frac{{}^j D}{{}^i D} \qquad . \tag{19}$$

Merlivat (1978) and later Luz et al. (2009) quantified the diffusivity ratios at 1.0251 and 1.0285 for $^1$H$^2$H$^{16}$O or $^1$H$_2^{18}$O
isotopologues, respectively. Dongmann et al. (1974) (but see also Brutsaert, 1975) extended Eq. (19) to different aerodynamic
regimes in the air boundary layer delimited by $z_{\text{EF}}$ and $z_{\text{atm}}$:

$$\alpha_K = \left(\frac{{}^j D}{{}^i D}\right)^n \tag{19'}$$

where $n$ [-] is an unitless factor ranging from 0.5 (corresponding to fully turbulent conditions) to 1 (fully diffusive), with a
value of 2/3 representative of laminar flow conditions. Mathieu and Bariac (1996) proposed to define $n$ in the case of
evaporation from soil as a linear function of soil volumetric water content observed in the surface layer ($\theta_{\text{surf}}$ [L$^3$ L$^{-3}$], typically
in cm$^3$ cm$^{-3}$). $n$ would range between 0.5 when $\theta_{\text{surf}}$ reaches $\theta_{\text{sat}}$, the water content value at saturation, and 1 for $\theta_{\text{surf}} = \theta_{\text{res}}$, the
value of residual water content (see Fig. 4):

$$n = \frac{(\theta_{\text{surf}}-\theta_{\text{res}})n_{\text{atm}}+(\theta_{\text{sat}}-\theta_{\text{surf}})n_{\text{soil}}}{\theta_{\text{sat}}-\theta_{\text{res}}} \tag{20}$$

In Mathieu and Bariac's conceptual framework the establishment of a dry soil surface layer results in added isotopic resistance
by increasing the relative importance of gaseous molecular diffusion (i.e., in the tortuous soil pores network) in the overall





transport of water vapour from the EF towards the well mixed atmosphere. In case of a fully water-saturated layer in contact with the free atmosphere, the opposite happens: water vapour leaving the rough surface is preferentially transported in a turbulent manner, leading to smaller $n$ values.

### 3.2.2 Progress and challenges

To calculate $\delta_E$ with the Craig and Gordon equation requires simultaneous observations of $h_{atm}$, $T_{EF}$, $\delta_{atm}$ and $\delta_{EF}$. The first two
variables are typically monitored with time domain reflectometry or capacitive sensing. As for $\delta_{atm}$, its value is determined from online or offline isotopic analysis after sampling of the atmospheric water vapour (see section 3.1).

The variable most challenging to estimate is $\delta_{EF}$ (Fig. 4b and e). It greatly depends on how soil is sampled at the EF. However, there is no consensus on how this should be done in the literature (see column "Isotopic measurements" of Table A2). Some studies do not precisely report the soil depth, which is considered to be the EF (e.g., Wang and Yakir, 2000; Yepez et al., 2003;
Williams et al., 2004). In others studies (Yepez et al., 2005; Zhang et al., 2011; Dubbert et al., 2013) soil profiles are partially or entirely sampled at higher vertical (cm) resolution. Pioneer works on isotopic transport in saturated/non-saturated isothermal soils under steady-state evaporation (Zimmermann et al., 1967; Allison, 1982; Barnes and Allison, 1983) showed that the EF, i.e., the theoretical and continuous boundary between the soil 'regions' dominated by either liquid or vapour flow (Fig. 4a and f), is associated with the highest isotopic composition ($\delta_{soil}^{liq}$) value of the liquid soil water (Fig. 4d-f). Later this family of
models was extended to unsaturated soil water conditions, non-isothermal conditions, and time-variable evaporation flux (e.g., Barnes and Allison, 1988; Barnes and Walker, 1989). More recently, Braud et al. (2005) and Haverd and Cuntz (2010) implemented isotopic transport in both liquid and vapour phases of the soil, with a coupling to temperature dynamics, in numerically solved SVAT models (SiSPAT-Isotope and Soil-Litter-Iso). All the above-mentioned studies underline the localized character of the EF and the strong isotopic gradient in liquid water at its location. The determination of the EF
location may be problematic, especially in the case of a receding EF ($z_{EF} \neq z_{surf}$, Fig 4d), which is generally the case in arid regions between rare precipitation events. Thus, sampling soil roughly from the surface does not allow for a precise determination of $\delta_{EF}$ and may lead to errors in $\delta_E$ estimates. Rothfuss et al. (2010) could demonstrate for a well-watered soil (i.e., with $z_{EF} = z_{surf}$, Fig 4b) that sampling of only a few cm of soil at the surface and using the corresponding $\delta_{surf}$ in Eq. (18) could lead to a significant underestimation of $\delta_E$. This would lead in turn to an overestimation of the T/ET ratio, since

negative changes in $\delta_E$ translate into positive changes in T/ET, i.e., $\frac{\partial\left(\frac{T}{ET}\right)}{\partial(\delta_E)} = \frac{\delta_{ET}-\delta_T}{(\delta_T-\delta_E)^2} < 0$ (when $\delta_{ET}<\delta_T$, which is generally the case). The spatial (vertical) resolution of the soil sampling should therefore be as high as possible to be able to identify $z_{EF}$ precisely. For their specific case, Brunel et al. 1997 estimated also that the determination of the $\delta_{EF}$ value was the greatest source of uncertainty of T/ET.

After sampling in the field, water is recovered from the soil in the laboratory using one of six extraction methods: cryogenic
vacuum extraction (Araguás-Araguás et al., 1995; West et al., 2006), azeotropic distillation (Revesz and Woods, 1990), direct vapour equilibration (Wassenaar et al., 2008), high pressure mechanical squeezing (Kelln et al., 2001), centrifugation



(Mubarak and Olsen, 1976), or microwave extraction (Munksgaard et al., 2014). Other methods include the use of soil liquid water samplers (Wenninger et al., 2010; Sutanto et al., 2012). Finally, $\delta_{EF}$ is measured by isotope ratio mass spectrometry (IRMS) or isotope ratio infrared spectrometry (IRIS). Note that an alternative consists in letting soil water directly equilibrate

with $CO_2$ without the need for water extraction (one study: Ferretti et al., 2003; after the method of Scrimgeour, 1995). In this framework, pure $CO_2$ is injected in the exetainer containing the soil sample following evacuation. After a three day-long water-$CO_2$ equilibration period, the $\delta^{18}O$ value of $CO_2$ is measured by isotope mass spectrometry and used to infer that of water at equilibrium. Orlowski et al. (2016a; 2016b) provided evidence from laboratory benchmarks of the different techniques that the isotopic composition of the recovered water could be sensitive to the extraction approach and extraction time as well as to

the soil type, and water and organic content values. The same authors also observed that IRMS and IRIS techniques yielded different results in general, and especially for clay loam soil water, which they related to interferences in the absorption spectrum during analysis with the latter technique. In addition, Orlowski et al. (2018) concluded from a worldwide inter-comparison of cryogenic vacuum extraction facilities that the general consensus in the isotopic ecohydrology community, stating that cryogenic vacuum extraction is the standard water recovery technique, should be questioned. Orlowski et al.

(2016a; 2016b; 2018) highlighted the limitations of the most popular extraction approach, i.e. based on the combination of destructive sampling and vacuum extraction (69 % of the reviewed studies), which calls for the development of other techniques for a precise quantification of $\delta_{EF}$.

In the last few years Rothfuss et al. (2013), Volkmann and Weiler (2014) and Gaj et al. (2016) successfully validated and tested alternatives to destructive sampling and offline isotopic analysis approaches. They developed systems based on the

combination of gas-permeable membranes (e.g., rigid hydrophobic microporous polypropylene, Membrana GmbH, Germany, or polyethylene, Porex Technologies, Aachen, Germany) with laser-based spectrometry for the non-destructive collection of the soil atmosphere and the online monitoring of its water vapour isotopic composition ($\delta_{soil}^{vap}$). For this, the soil atmosphere is either

(i)        flushed with a carrier gas (dry synthetic air, i.e., 20.5% in $N_2$, or 100 % $N_2$) at low flow rate in the range of 50-100

510            ml min$^{-1}$ (Rothfuss et al., 2013; Volkmann and Weiler, 2014; Gaj et al., 2016) or

(ii)       extracted with a vacuum pump (Volkmann and Weiler, 2014).

Both modi operandi allow for long-term and repeated measurements across the soil profile provided that condensation is avoided in the sampling line. For this, the collected air, which is (quasi-)saturated with water vapour, is diluted with the carrier gas and the sampling lines are heated, if necessary (Quade et al., 2019; Kühnhammer et al., 2019). Rothfuss et al. (2013)

observed near-isotopic equilibrium conditions between liquid and vapour in the soil pore space, and provided temperature calibration equations yielding results analogous to those of Majoube (1971) and Horita and Wesolowski (1994) for converting $\delta_{soil}^{vap}$ into $\delta_{soil}^{liq}$ values. They further show that isotopic equilibrium conditions still prevailed at low soil volumetric water content, possibly also for soil water vapour relative humidity values lower than one. Their method was successfully applied to laboratory experiments with sand (Gangi et al., 2015; Rothfuss et al., 2015) and silt loam (Quade et al., 2018). Oerter et al.





(2017) compared $\delta_{\text{soil}}^{\text{liq}}$ values estimated with the monitoring method of Rothfuss et al. (2013) on the one hand, and the direct

equilibrium and vacuum extraction methods on the other hand. They found a good correlation between the two approaches

(root mean square error – RMSE equal to 0.6 ‰ for $\delta^{18}$O and within 1.7-3.1 ‰ for $\delta^2$H). Volkmann and Weiler (2014) tested

their own design of a water vapour probe under field conditions and could show that it produced $\delta_{\text{soil}}^{\text{liq}}$ values in agreement with

those following destructive sampling and isotopic analysis with the direct equilibration method (Garvelmann et al., 2012). The

inter-method (destructive vs. non-destructive) RMSE values were comparable to the intra-method variability of soil water $\delta$-

values. The latter variability could not be disentangled into systematic methodological error and natural (lateral) heterogeneity

in soil water isotopic composition. Kübert et al. (2020) conducted a comparison study of the method of Rothfuss et al. (2013)

with cryogenic vacuum extraction and centrifugation during an irrigation pulse-labelling experiment in a semi-natural

temperate grassland. They highlighted that the non-destructive method could capture temporal dynamics of the isotopic

composition, while destructive sampling included both the temporal change and spatial heterogeneity.

To date there are two ET partitioning studies, in which $\delta_E$ was determined from non-destructive isotopic analysis using soil

liquid water-water vapour equilibration. Quade et al. (2019) applied the method of Rothfuss et al. (2013) in a sugar-beet field

in a temperate climate (Germany), while Gaj et al. (2016) used commercially available soil gas probes (BGL-30, METER

Group, Munich, Germany), following the same modus operandi as Volkmann and Weiler (2014), during a field study in central

Namibia under semi-arid conditions. Such applications are promising for the specific purpose of partitioning ET, as they

provide insights into sub-daily dynamics of $\delta_E$ from the online assessment of the positioning and isotopic composition of water

at the EF. However, one noticeable disadvantage is the need for deploying a laser spectrometer at the experimental site. A

possible way around has been lately proposed by Havranek et al. (2020) as a compromise: water vapour samples are collected

and stored automatically in flasks from the soil profile in the field following the approach of Rothfuss et al. (2013) and

transported back to the laboratory where the isotopic analyses are performed.

Another important factor that influences the precision of $\delta_E$ estimates is the choice of the value of the kinetic fractionation

factor $\alpha_K$. Only a handful of studies attempted to estimate or model $\alpha_K$ for soil E. Braud et al. (2009) simulated $\alpha_K$ values during

long-term laboratory experiments with the SVAT model SiSPAT-Isotope. They found a decreasing trend in $\alpha_K$ value from

saturated to unsaturated soil conditions, which contradicts the model of Mathieu and Bariac (1996). Results similar to the study

by Braud et al. (2009a) were obtained by Rothfuss et al. (2015) during a long-term soil column laboratory experiment. Quade

et al. (2018) tested two different methods for quantifying $\alpha_K$ during a series of bare soil evaporation experiments on monoliths

(100 L soil volume) under semi-controlled conditions, i.e.,

(i)        by inversion of the Craig and Gordon equation (Eq. (18)) in a single isotope-framework (i.e., based on either $\delta^{18}$O or

            $\delta^2$H values) with input variables $h_{\text{atm}}$, $T_{\text{EF}}$, $\delta_{\text{atm}}$, $\delta_{\text{EF}}$, and $\delta_E$;

(ii)      by inversion of the Craig and Gordon equation in a dual isotope-framework. More specifically, $\alpha_K$ is determined from

            the approximation of the slope of the soil E line ($S_E[-] = \Delta\delta^2\text{H}_{\text{soil}}^{\text{liq}} / \Delta\delta^{18}\text{O}_{\text{soil}}^{\text{liq}}$) in a [$\delta^{18}$O, $\delta^2$H] coordinate system

            following Gat (2000):





$$S_E(t) = \frac{\left[h(t)\left(\delta_{atm}(t) - \delta_{EF}(t-1)\right) + \varepsilon_{eq}(t) + \Delta\varepsilon(t)\right]_{2_H}}{\left[h(t)\left(\delta_{atm}(t) - \delta_{EF}(t-1)\right) + \varepsilon_{eq}(t) + \Delta\varepsilon(t)\right]_{18_O}},$$

(21)

with $t$ the time stamp and $\Delta\varepsilon$ [-], the so-called kinetic effect, is defined as

$$\Delta\varepsilon = (1 - h_{atm})(\,^{j}D/\,^{i}D - 1)n.$$

(22)

with superscripts $i$ and $j$ standing for the least and most abundant isotopologues, respectively. Equation (21) is solved in an implicit manner, in other words, $S_E$ values simulated for time stamp $t$ depend on $\delta_{EF}$ observation made at time stamp $(t-1)$. The $n$ value is then extracted from Eq. (21) from the confrontation of measured and simulated $S_E$, and finally fed into Eq. (19') to retrieve $\alpha_K$ values. Quade et al. (2018) showed that $\alpha_K$ could not be considered as a constant value solely depending on flow

conditions as proposed by Dongmann et al. (1974) or determined from soil water content following Mathieu and Bariac (1996) (Eq. (20)). The second approach yielded $\alpha_K$ values in agreement with the literature (e.g., Merlivat, 1979). Quade et al. (2018) concluded that turbulent transport still played a significant role during the evaporation process, also under non-saturated conditions. These studies show that further sensitivity analyses of $\alpha_K$ to environmental conditions are needed to provide realistic estimates of $\delta_E$ and ultimately of T/ET ratios. To our knowledge, there is no ET partitioning study in the field where

$\alpha_K$ was considered to dynamically change (other than via the model of Mathieu and Bariac, 1996) depending on the contribution of air turbulence to water vapour transport in the free and canopy atmosphere, e.g., from measurements of the wind profile within and above the canopy (Brutsaert, 1975).

Another source of uncertainty arises from a lack of precise knowledge of the state of water vapour saturation at the EF. In the Craig and Gordon equation, the kinetic fractionation factor is weighed by the term ($h_{EF} - h_{atm}$) where $h_{EF}$ is generally assumed

equal to 1, representative of saturated conditions at the EF. However, this assumption may not stand for dry soils considering the relationship between soil water matric potential $\psi_{EF}$ [M L$^{-1}$ T$^{-2}$, typically expressed in hPa or cm water height] and pore space relative humidity at the EF ($h_{EF}$), as given by the Kelvin law:

$$h_{EF} = \exp\left(\frac{\psi_{EF} M_w}{\rho_w R_{gas} T_{EF}}\right).$$

(23)

$M_w$ and $\rho_w$ [M L$^{-3}$] are the molar and volumetric masses of water, respectively, and $R_{gas}$ [M L$^{-1}$ T$^{-3}$] the universal gas constant.

Table 1 presents three different degrees of saturation of the soil vapour phase under isothermal conditions ($T_{EF} = 20°C$) and their corresponding hydrogen and oxygen isotopic composition values of the E flux ($\delta^2H_E$ and $\delta^{18}O_E$). A decrease in $h_{EF}$ from 100 to 99.9 %, corresponding to an increase in the absolute $\psi_{EF}$ value from 0 to 1000 hPa (i.e., from saturation to pF=3) leads, for example, to an increase of 1.5 ‰ in $\delta^2H_E$ and $\delta^{18}O_E$. A decrease in $h_{EF}$ from 100 to 99.3 (increase from 0 to 10000 hPa, i.e. pF=4) would translate into an increase of 13 ‰ in $\delta^2H_E$ and $\delta^{18}O_E$. Both $\delta^2H_E$ and $\delta^{18}O_E$ are affected in the same way by the

change in value of the factor $\frac{1}{\alpha_K(h_{EF} - h_{atm})}$ (see Eq. (18)), i.e., approximatively 2.0‰ per 0.1% relative humidity. This may have a noticeable effect on the computation of the T/ET ratio, especially for $\delta^{18}O$, for which the difference $\delta_T - \delta_E$ is usually smaller than for $\delta^2H$.





### 3.3 Isotopic composition of transpiration

#### 3.3.1 Methods

The determination of the isotopic composition of T, $\delta_T$, in the reviewed literature is mainly depending on the underlying hypothesis on isotopic steady or non-steady state (NSS) of T. While 42 % of all reviewed studies assume isotopic steady state (ISS), in other words that $\delta_T$ is time-invariant, 58 % do not make such an assumption but assume a transient state, i.e. NSS (Fig. 1j). This has substantial implications for the materials and methods used for the determination of $\delta_T$. In the ISS approach, $\delta_T$ is directly inferred from the isotopic value of the leaf water source ($\delta_{xyl}$), i.e., the water in the xylem vessels supplying the

leaf water reservoir. This assumption is based the fact that at ISS the flux density rate of the least abundant ($^iF_{xyl}$) (respectively most abundant, $^jF_{xyl}$) isotopologue entering the leaf equals the flux density rate of the least abundant ($^iF_T$) (most abundant, $^jF_T$) isotopologue leaving it by transpiration:

$$^jF_{xyl} = {}^jF_T, \tag{24a}$$

$$^iF_{xyl} = {}^iF_T \Rightarrow {}^jF_{xyl}\delta_{xyl} = {}^iF_T\delta_T \Rightarrow \delta_{xyl} = \delta_T. \tag{24b}$$

Note that in this framework an instantaneous change in $^jF_T$, if compensated by a corresponding change in $^jF_{xyl}$, should maintain the relationship $\delta_{xyl} = \delta_T$ (Eq. (24b)). In reality, a change in $^jF_T$, due to variations in environmental factors (e.g., vapour pressure deficit of the free atmosphere and incoming solar radiation) implies a change in root water uptake depth profile, which in turn affects $\delta_{xyl}$ in case of a heterogeneous distribution of the soil water isotopic composition (Rothfuss and Javaux, 2017). A new ISS is eventually reached, depending on the leaf water turnover time, i.e. the ratio of leaf water volume

and transpiration rate (Dongmann et al., 1974; see below). To access xylem water, authors destructively sample stems (e.g., Wei et al., 2018; Quade et al., 2019), branches (e.g., Williams et al., 2004), or root water (Bijoor et al., 2011), and recover their water by, e.g., cryogenic vacuum extraction.

The NSS approach for determining $\delta_T$ relies either on direct non-destructive monitoring (i.e., leaf chamber-based measurements, e.g., Wang et al., 2010) or on destructive sampling of plant material and subsequent extraction of water (e.g.,

Dubbert et al., 2013). In the former case, the modus operandi is the same as when operating ET and E chambers coupled to mass balance equations (see sections 3.1 and 3.2, respectively), except that one single leaf or several leaves are enclosed in the chamber (with a volume ranging from 150 to 190 cm$^3$ in the literature), rather than the entire plant. It is then generally assumed that the leaf-scale $\delta_T$ estimate is also representative for the whole plant (e.g., Good et al., 2014). In the case of destructive sampling, $\delta_T$ is modelled on the basis of environmental factors (leaf temperature and free atmosphere relative humidity) and

isotopic variables. Two cases can be distinguished:

(i)    $\delta_T$ is determined from the value of the isotopic composition of the leaf bulk water, $\delta_L$, with the Craig and Gordon equation adapted to plant T (Sun et al., 2014; Hu et al., 2014):



$$\delta_T = \frac{1}{\alpha_K(1-h)}\left(\frac{\delta_L+1}{\alpha_{eq}} - (\delta_{atm}+1)h\right) - 1; \tag{18'}$$

(ii)  the isotopic composition of leaf water may not be available, but that of its source, $\delta_{xyl}$. The $\delta_T$ value is calculated after
the relationship of Dongmann et al. (1974), which describes the temporal course of $\delta_L$ at constant transpiration rate
value (i.e., at permanent flow for T). The authors expressed the rate of change in $\delta_L$ as a function of the instantaneous
difference between $\delta_{xyl}$ and $\delta_T$ at time $t$, by considering the leaf bulk water (delimited by volume per unit leaf area $V_L$
[L$^3$ L$^{-2}$]) to be transpired into ambient air at permanent flow (i.e., at density rate $^jF_T = {}^jF_{xyl}$, as in Eq. (24a)):

$$d\delta_L = \frac{^jF_T}{V_L}\left(\delta_{xyl}(t) - \delta_T(t)\right)dt \tag{25}$$

By combining Eqs. (18') and (25) and considering that $\delta_{xyl}$ is time-invariant, we obtain a first-order differential equation for
$\delta_L$, which yields after integration to:

$$\delta_L(t) = \delta_L(t\to+\infty) - \left(\delta_{xyl} - \delta_L(t\to+\infty)\right)\exp\left(-\frac{t}{\tau_L}\frac{1}{\alpha_{eq}\alpha_K(1-h_{atm})}\right) \tag{26}$$

where the leaf water turnover time, $\tau_L$, is defined as the ratio $\frac{V_L}{^jF_T}$ and $\delta_L(t\to+\infty) = \delta_{L\_ISS}$, the isotopic composition of leaf
bulk water when an isotopic steady state is reached. The latter term is expressed as:

$$\delta_{L\_ISS} = \alpha_{eq}\left[\alpha_K(1-h)(\delta_{xyl}+1) + h_{atm}(\delta_{atm}+1)\right] - 1. \tag{27}$$

By (i) noting $\alpha_{eq} = \varepsilon_{eq}+1$ and $\alpha_K = \varepsilon_K+1$, where $\varepsilon_{eq}$ and $\varepsilon_K$ [-] are the equilibrium and kinetic fractionations, respectively,
and (ii) dropping terms with products $\varepsilon_{eq}\times\varepsilon_K$, we obtain the following expression of the difference in isotopic composition
between leaf and source waters at ISS:

$$\delta_{L\_ISS} - \delta_{xyl} = \varepsilon_{eq} + \varepsilon_K + h_{atm}\left(\delta_{atm} - \delta_{xyl} - \varepsilon_K\right) \tag{27'}$$

We note that Eq. (27') is the inversion of the Craig and Gordon equation at ISS, i.e., when $\delta_T = \delta_{xyl}$. Finally, $\delta_T$ is computed
with the NSS-Craig and Gordon equation, i.e., Eq. (18'). Eq. (26) states that, at permanent state for transpiration, the degree
of attainment of ISS conditions in the leaf is a function of time, leaf internal dynamics ($\tau_L$) and (isotopic) aerodynamic boundary
conditions. The formula of Dongmann et al. (1974) requires two additional parameters as compared to the more
'straightforward' application of the Craig and Gordon equation, namely leaf transpiration ($^jF_T$) and volume ($V_L$), both labour-
intensive to obtain and associated with high uncertainties.

Both case scenarios (i) and (ii) make the assumption that leaf water is a well-mixed reservoir, in other words that only
convective transport of the water isotopologues occurs, leading to $\delta_L = \delta_{Lts}$, where $\delta_{Lts}$ is the isotopic composition of water at
the leaf transpiration sites. However, a number of studies reported strong isotopic variations within the leaf water pool (i.e.,
among different compartments such as leaf veins, cell walls, and symplastic water, see e.g., Yakir et al., 1989; Wang et al.,
1998; 1994; Bariac et al., 1994), which can be related to hydraulic separation of water pools and diffusive transport from the





transpiration sites towards the petiole of the leaf. Another explanation may be found in the heterogeneity in opening of the leaf stomata (Farquhar et al., 2007). More specifically, $\delta_{Lts}$ should be significantly higher than the bulk leaf water isotopic composition value $\delta_L$, which leads to an underestimation of $\delta_T$ by the direct application of the Craig and Gordon equation. Walker et al. (1989), Walker and Brunel (1990), and Flanagan et al. (1991) considered in a first approach two distinct water

pools in the leaf, one in isotopic equilibrium with water vapour in the stomatal cavity (of isotopic composition $\delta_{Lts\_ISS}$) and one isotopically undistinguishable from xylem water (of isotopic composition $\delta_{xyl}$) in respective proportions $p$ and $(1–p)$. In these three studies, an analogous expression to Eq. (27') is used where $p$ is accounted for:

$$\delta_{Lts\_ISS} - \delta_{xyl} = \frac{\delta_{L\_ISS} - \delta_{xyl}}{p} = \varepsilon_{eq} + \varepsilon_K + h_{atm}\left(\delta_{atm} - \delta_{xyl} - \varepsilon_K\right) \tag{27''}$$

They suggested that there was a midday maximum for T density rate from the corresponding minimum value for $p$. Cernusak

et al. (2002) and Farquhar and Cernusak (2005) proposed a similar equation to that of Dongmann et al. (1974) for the evaporative isotopic enrichment in leaves in NSS conditions, but without considering the leaf water volume per unit area constant in time. Eq. (25) becomes in their case:

$$d(V_L\delta_L) = {}^jF_T\left(\delta_{xyl}(t) - \delta_T(t)\right)dt. \tag{25'}$$

By replacing $\delta_{xyl}$ and $\delta_T$ in the right hand-term of Eq. (25') by the ISS- and NSS-Craig and Gordon equation forms, respectively,

the authors give an expression relating the rate of change of $\delta_L$ with the difference between $\delta_{Lts\_ISS}$ and $\delta_{Lts}$:

$$\frac{d(V_L\delta_L)}{dt} = \frac{{}^j\chi_{int}}{{}^jr \times \alpha_K\alpha_{eq}}\left(\delta_{Lts\_ISS} - \delta_{Lts}\right), \tag{28}$$

where ${}^j\chi_{int}$ and ${}^jr$ are the water vapour mixing ratio in the intercellular space and, as in section 3.2, the resistance to vapour flow of the ${}^1H_2{}^{16}O$ isotopologue in air, respectively. It is therefore possible, by fitting the time course of the bulk leaf water isotopic composition $\delta_L$ to deduce $\delta_{Lts}$, on the basis of which $\delta_T$ is finally determined using Eq. (18') (Yepez et al., 2005). $\alpha_{eq}$ is, as in

section 3.2, calculated following the closed-form equations of, e.g., Horita and Wesolowski (1994) (Eq. (17)). As for $\alpha_K$, its expression is adapted to include the series of flow resistances of water vapour isotopologues inside the stomatal cavity/through the stomatal opening (${}^ir_{sto}$ and ${}^jr_{sto}$ [T L$^{-1}$]) and in the leaf boundary layer (${}^ir_{bdl}$ and ${}^jr_{bdl}$ [T L$^{-1}$]) (Jarvis, 1976; Stewart, 1988). Farquhar et al. (1989) (and see also Cernusak et al., 2005; Farquhar et al., 2007) considered that molecular diffusion drives the transport of the different water vapour isotopologues in the first case, and that turbulence prevails in the second, leading to $n$

exponent values of 1 and ½, respectively (Dongmann et al., 1974; Eq. (19')). In this framework, $\alpha_K$ is decomposed as:

$$\alpha_K = \frac{{}^ir}{{}^jr} = \frac{{}^ir_{sto} + {}^ir_{bdl}}{{}^jr_{sto} + {}^jr_{bdl}} = \frac{\left(\frac{{}^jD}{{}^iD}\right)^1 \cdot {}^jr_{sto} + \left(\frac{{}^jD}{{}^iD}\right)^{1/2} \cdot {}^jr_{bdl}}{{}^jr_{sto} + {}^jr_{bdl}} \tag{29}$$




Cuntz et al. (2007) proposed a general iterative solution of Dongmann et al. (1974)'s formulation revisited by Cernusak et al. (2002) (Eq. (28)) under various scenarios depending on considerations regarding leaf water reservoir isotopic homogeneity ($\delta_L = \delta_{Lts}$ or $\delta_L \neq \delta_{Lts}$) and volume ($dV_L/dt = 0$ or $dV_L/dt \neq 0$). Dubbert et al. (2013) applied their solution in the case of an isotopically well-mixed leaf water pool transpiring at constant volume, and expressed the incremental change in $\delta_L$ from time step $t$ to $t+dt$ as:

$$\delta_L(t + dt) = \delta_{L\_ISS} + \left(\delta_L(t) - \delta_{L\_ISS}\right) \exp\left(-\frac{g_s{}^j \chi_{int}}{\alpha_K \alpha_{eq} V_L} dt\right) \tag{30}$$

where $g_s$ [L T$^{-1}$] is the total stomatal conductance.

### 3.3.2 Progress and challenges

The isotopic composition of T may be derived under NSS conditions from plant chamber measurements following Eq. (7) (section 3.1), either at the leaf level or at the branch level. While most studies developed and operated custom-made chambers, only few (e.g., Wang et al., 2010) used commercially available leaf chambers (e.g., LICOR-6400, Nebraska, USA). Chamber measurements have several disadvantages as discussed in section 3.1, but are essential for monitoring $\delta_T$ directly without relying on additional modelling steps using either $\delta_{xyl}$ or $\delta_L$, the determination of which, i.e., based on destructive sampling and water recovery with, e.g., cryogenic vacuum extraction, is also associated with uncertainty (e.g., Orlowski et al., 2016a; 2016b; Millar et al., 2018). The important two features of the chamber-based method are that it does not require the assumption of ISS, and that it allows for repeated (i.e., non-destructive) measurements on the same leaf or ensemble of leaves during the course of the day.

A novel type of non-destructive method, first published by Volkmann et al. (2016) and lately by Marshall et al. (2020), could enable monitoring $\delta_T$ of trees at an equivalent temporal resolution and even greater temporal coverage than with leaf- or plant-scale chamber systems. In the former study, several 10-mm outer diameter gas probes designed after Volkmann and Weiler (2014) (see section 3.2) were inserted into pre-drilled holes in the trunk sapwood of two individuals of *Acer campestre* L. The probes were positioned at breast-height in various azimuths. By assuming isotopic equilibrium between the water vapour sampled by the probe and flushed to the laser spectrometer and the xylem (liquid) water, the authors computed $\delta_{xyl}$ values from the temperature-dependent relationships given by, e.g., Majoube (1971) and Horita and Wesolowski (1994). For comparison, tree sapwood was destructively sampled and its water isotopic composition measured with IRMS after cryogenic vacuum extraction. A good agreement was found between online measurements and offline analysis of xylem water hydrogen isotopic composition. The inter-method bias regarding the determination of xylem water $\delta^{18}O$ was thought to be due to spectral interferences during online analysis with the laser spectrometer. The experimental natural conditions did not allow the authors to conclude if differences in $\delta_{xyl}$ among the different gas probes reflected actual diurnal variations in root water uptake or preferential connection between xylem vessels and specific parts of the root system that were not affected by the labelling pulse. The authors underline the difficulty with their experimental design to precisely measure the temperature of equilibration



in the gas probe (needed for converting sample water vapour to xylem water isotopic composition), due to the high lateral
temperature gradient and its daily course. Marshall et al. (2020) tested a cruder way (which they entitled the "Borehole
Equilibration") to sample water vapour originating from xylem water of two pine tree species (*Pinus sylvestris* L. and *Pinus pinea* L.) under semi-controlled conditions. Contrary to Volkmann et al. (2016), the authors (i) did not use a gas probe but
simply connected a hole drilled horizontally through the trunk to a laser spectrometer with gas sampling lines. Furthermore,
(ii) the experiments were performed in hydroponic water solutions to enable a quasi-instantaneous change of the isotopic
composition of the water source, thereby setting defined lower isotopic boundary conditions for further modelling efforts. To
test the practicability of the method, the experimental results were confronted with a 'Dongmann-like' NSS formulation of the
isotopic composition of the water vapour stream, in which the geometry and its consequence on the diffusion from the borehole
surface and on the establishment of laminar flow transport were explicitly accounted for. With their model, the authors tested
whether the sampled water vapour was in isotopic equilibrium with xylem water or was the product of evaporation from it. It
was shown that the prevalence of a full isotopic equilibrium was a reasonable assumption and that the flow-through time (i.e.,
borehole volume divided by the flow rate) was 20 times greater than the time needed for diffusion of water vapour originating
from the xylem vessels into the laminar flow region in the middle of the borehole section. Both methods present a drastic
advancement in isotopic analysis of xylem water and have great potential in the context of ET partitioning of forest ecosystems,
on the pivotal condition that the steady state assumption ($\delta_{xyl} = \delta_T$) applies during periods of measurements. The long-term
applicability of the method, i.e., the ability of the investigated tree species to withstand the invasive and destructive installation
of the probe, still needs to be proven at this point.

While the coupling between gas-exchange chambers and laser spectrometers has the advantage of directly measuring $\delta_T$, the
aforementioned destructive sampling method and in-situ monitoring technique quantify $\delta_L$ or $\delta_{xyl}$, therefore may require a
modelling step to obtain $\delta_T$. While a number of studies (e.g., Zhou et al., 2018; Wei et al., 2015; Aouade et al., 2016; Volkmann
et al., 2016) assume ISS and hence argue that $\delta_{xyl}$ equals $\delta_T$, there is growing evidence that plants rarely reach ISS throughout
the day (Simonin et al., 2013; Dubbert et al., 2014b; 2017). Moreover, the leaf water turnover time, which can effectively be
described by stomatal conductance ($g_s$), vapour pressure deficit and leaf water volume, is species-specific and ranges from
several minutes to several hours (Song et al., 2015). As the leaf water turnover time describes the necessary time for leaf water
to reach ISS (see exponent terms in Eqs. (26) and (30)), ISS can either be observed for large parts of the day (e.g., in many
herbaceous species) or not at all (e.g., in plant species strongly controlling their $g_s$, see Dubbert et al. (2017) and Dubbert and
Werner (2019) for an overview). Therefore, the validity of assuming ISS for the purpose of ET partitioning will largely depend
on the desired temporal scale: considering NSS is definitely necessary at sub-diurnal to diurnal scale, but unimportant at larger
time scales. In case NSS is likely to occur, $\delta_T$ can be modelled using a 'Dongmann version' of the Craig and Gordon equation,
as shown in the previous sub-section 3.3.1 (Dongmann et al., 1974). However, this complicates the partitioning approach
considerably in comparison to direct chamber measurements of $\delta_T$, as a large number of additional observations are necessary.
In particular, $g_s$ and the canopy temperature are important input parameters. Therefore, the use of chamber measurements is
highly recommended in any case.





The choice of an appropriate method for measuring the isotopic composition of unfractionated xylem water is crucial for a correct determination of $\delta_T$. For example, herbaceous, grass or crop species do not have suberized stems, thus destructive sampling would have to rely on leaf water sampling or sampling the plant culm belowground, which is highly destructive and
not possible on plots of common size. Moreover, while the majority of studies still provide evidence for an unfractionated root uptake and transport of xylem water through plants, there is growing evidence of fractionation of xylem water during times with low transpiration rate (drought condition, see e.g., Martin-Gomez et al., 2017) for deciduous species.

## 4 Possible ways forward

The isotopic methodology for partitioning ET relies on a number of possible combinations of different techniques, which differ
in numerous aspects. While some of them are based on destructive sampling and water recovery using one of the aforementioned methods (e.g., cryogenic vacuum extraction, direct liquid-vapour equilibration, see section 3.2) and *a posteriori* analysis in the laboratory (e.g., for determination of $\delta_E$ using the Craig and Gordon equation), other methods are non-destructive, provide online measurements and do not include a strong modelling component (e.g., determination of $\delta_T$ with plant chambers with one of two mass-balance techniques). Destructive approaches do not require the handling of soil,
plant, or soil & plant chambers, nor the deployment of a laser spectrometer along with its conditioning system in the field. They should also allow for capturing the inherent spatial variability with repeated sampling (however, at the cost of long hours spent in sample preparation and water extraction). Non-destructive methods, such as chambers, may on the other hand provide environmental conditions for the enclosed plant that are not representative of ambient conditions.

Up to now only indirect methods, e.g., based on Scanlon and Kustas (2010), might be able to provide continuous and sub-daily
estimates of T/ET. Some methods, such as the Keeling plot technique, can provide long-term continuous estimates of $\delta_{ET}$ once a meteorological mast is installed in the field. It is, on the other hand not advisable to enclose a plant in a chamber over longer time periods. Within the realm of destructive techniques, the user may assume ISS or test its existence when determining the isotopic composition of T. The techniques, with which $\delta_{ET}$ is estimated generally differ in terms of spatial significance as compared to those for determining $\delta_E$ and $\delta_T$. Estimates of $\delta_{ET}$ obtained either with the EC, Keeling plot, or flux-gradient
technique are thought to be representative at the field scale (e.g., as represented by the EC footprint). Note that this is also a problem encountered in (non-isotopic) instrumental approaches for partitioning ET, including EC, micro-lysimeters and soil chambers (Kool et al., 2014). To account for these discrepancies in spatial representativeness, several micro-lysimeters and (if possible automated) chambers are deployed on site, e.g., within the framework of global networks (e.g., FLUXNET; Law et al., 2002). On the contrary, there is no consensus to date on a common methodological ground for partitioning ET in the field
on the basis of water stable isotopic measurements, depending on the type of land cover and use (agricultural, grassland or forest ecosystems).



It is the authors' belief that non-destructive and online methods integrated into automated sampling platforms and part of long-term (e.g., multi-year) water flux observatories should be preferred over destructive and punctual assessments of T/ET ratios. In this (ideal) framework, we propose that

(i)       the seminal effort in applying the EC technique by Griffis et al. (2010) should be continued to provide half-hourly and continuous ecosystem-scale $\delta_{ET}$ estimates. The $\delta_{ET}$ estimates obtained with the EC technique should be corroborated/confronted with the Keeling plot and the flux-gradient approaches to identify possible scale-dependent disparities in surface isotopic signals as in Good et al. (2012);

        (ii)      $\delta_E$ should be monitored by installing gas-permeable membranes or tubing (see section 3.2) in the upper layers of the
770               soil, depending on site-specific knowledge regarding the receding of the EF. While the gas probes of Volkmann and Weiler (2014) and Gaj et al. (2016) are better-suited for insertion at different locations in a soil profile, the membrane tubing used by Rothfuss et al. (2013), Oerter and Bowen (2019) and Kübert et al. (2020) allow to cover more ground surface by using a customized length of tubing. This should help to increase the representativeness of the $\delta_E$ value estimated from the soil water vapour isotopic composition and the use of the Craig and Gordon equation. When using
775               the model of Craig and Gordon (1965), authors should systematically perform sensitivity analyses of

                  a.    the depth of the EF and its water isotopic composition, and

                  b.    the value of the kinetic fractionation factor, $\alpha_K$.

        These analyses will provide insights into the uncertainty of the T/ET ratio, in addition to the uncertainty originating from the solution of the two end-member equation (Eq. (1)) (Rothfuss et al., 2010). This is, however, under-
investigated according to our literature review. The $\alpha_K$ value may be derived in a dual-isotope space using the formulation of Gat (2000), rather than based on unclear assumptions regarding the type of transport (molecular diffusion, laminar or turbulent transport) controlling the flow of water stable isotopologues (see section 3.2). As a side note (and without a proof of concept for this), the $\delta_E$ value may be directly determined in the case of a well-developed dry surface on the basis of non-destructive measurements of the soil water vapour isotopic composition
($\delta_{soil}^{vap}$) at two depths ($z_1$ and $z_2$) located between the EF and the soil surface. For this, the Craig and Gordon equation may be used without the need to locate the soil EF nor to assume liquid-vapour equilibrium:

$$\delta_E = \frac{\delta_{soil}^{vap}(z_1) \cdot h(z_1) - \delta_{soil}^{vap}(z_2) \cdot h(z_2)}{\frac{j_D}{i_D}(h(z_1) - h(z_2))} - 1; \tag{18''}$$

        (iii)     several transparent flushed plant-size chambers should be operated at the study site to characterize the *in situ* natural lateral heterogeneity of $\delta_T$, due to differences in root water uptake, plant physiological state, as well as lateral
790               heterogeneity in soil water isotopic composition profiles. Developments should be made towards designing chambers able to mimic the dynamic states of ambient air (temperature and relative humidity, wind turbulence) to avoid biases in $\delta_T$ estimation. This could be done by cooling of the inlet air to avoid over-heating of the air inside the chamber, and an adaptive active ventilation system. In situations where parts of the field are bare, e.g., between crop rows, soil



chambers should be installed as well to evaluate differences in $\delta_E$ between areas covered or not covered with

vegetation;

(iv)     the methods for monitoring of $\delta_{xyl}$ and its potential use in determining $\delta_T$ (that is, by assuming ISS conditions) have been tested and validated with tree species exclusively. The same principle is yet to be minimized and applied to crops able to survive the installation and carry the instrumentation, such as a well-developed maize plant.

Lastly, the lift system principle, as operated by Noone et al. (2013), Mayer et al. (2009), and recently for agricultural crops by

Ney and Graf (2018) has the potential to provide half-hourly concomitant values of $\delta_{ET}$, $\delta_T$ , and $\delta_E$ in the field. The principle is illustrated in Fig. 5, further developing that of Yepez et al. (2003). The Keeling plot technique is applied to data collected at high vertical resolution (ultimately implying high-frequency data acquisition of the analyser, typically equal or greater than 5Hz, see Ney and Graf, 2018) in three distinct atmospheric regions, i.e. (i) the region spreading from the fully turbulent atmosphere to the canopy height, (ii) the region comprised between canopy height (here fixed at 1.25 m) and the local maximum in $\delta_{atm}$, and (iii) the region delimited by the $\delta_{atm}$ local maximum and the ground level (Fig 5a). The y-intercepts of

the three Keeling plots give the concomitant values of the isotopic compositions of ET (Fig. 5b), T (Fig. 5c) and E (Fig. 5d). In this synthetic experiment, which cannot be construed as a proof of concept, $\delta^{18}O_{ET}$, $\delta^{18}O_T$, and $\delta^{18}O_E$ are equal to –4.7 (±1.5), –0.7 (±1.4), and –18.5 (±0.4) ‰, respectively, corresponding to a T/ET of 77 (±10) %.

## 5 Conclusion

Water stable isotopes are often described in the present literature compilation as "powerful" (or "insightful") tools for separating evaporation and transpiration fluxes. However, the number of ET partitioning studies, which the authors listed here, remains low when compared to the number of publications utilizing water stable isotopes for, e.g., determining plant water use strategies (30 versus 158 over the period 1990-2016, see Rothfuss and Javaux, 2017). The apparent contrast between the announced potential and the number of study cases is explained partly by both the complexity and multifaceted character of

the isotopic methodology. Unfortunately, and despite great efforts of the researchers, the spatial representativeness as well as temporal extent of the obtained T/ET data series are usually not well comparable with those of other non-isotopic methods (see Figure 1g).

The authors believe that, while ultimately increasing the complexity in terms of modus operandi, novel non-destructive monitoring methods are key to providing long-term T/ET data at the plot to the field scale and to upscaling local process

understanding to address large-scale ecohydrological issues in a changing climate.





## 6 Tables

| variables→ | $T_{EF}$ | $h_{atm}$ | $h_{EF}$ | $\alpha_K$ | | $\delta_{EF}$ | | $\delta_{atm}$ | | $\delta_E$ | |
|---|---|---|---|---|---|---|---|---|---|---|---|
| | [°C] | [%] | [%] | [-] | | [‰] | | [‰] | | [‰] | |
| isotopes→ | | | | $^2$H | $^{18}$O | $^2$H | $^{18}$O | $^2$H | $^{18}$O | $^2$H | $^{18}$O |
| ↓soil vapour phase state | | | | | | | | | | | |
| saturated | | | 100 | | | | | | | −32.6 | −23.2 |
| unsaturated [pF=3] | 20 | 50 | 99.9 | 1.0251 | 1.0285 | −4 | +2 | −120 | −20 | −31.1 | −21.7 |
| unsaturated [pF =4] | | | 99.3 | | | | | | | −18.1 | −8.6 |

**Table 1. Effect of the consideration of non-saturated soil water vapour phase on the estimation of the isotopic composition of evaporation ($\delta_E$) using the model of Craig and Gordon (1965). Conditions of pure diffusive water vapour transport (n=1) prevail, leading to values of the kinetic fractionation factor ($\alpha_K$) of 1.0251 und 1.0285 for $^2$H and $^{18}$O. Values for $T_{EF}$, $h_{atm}$, $\delta_{EF}$ and $\delta_{atm}$ are**
**chosen exemplarily.**





# 7 Figures

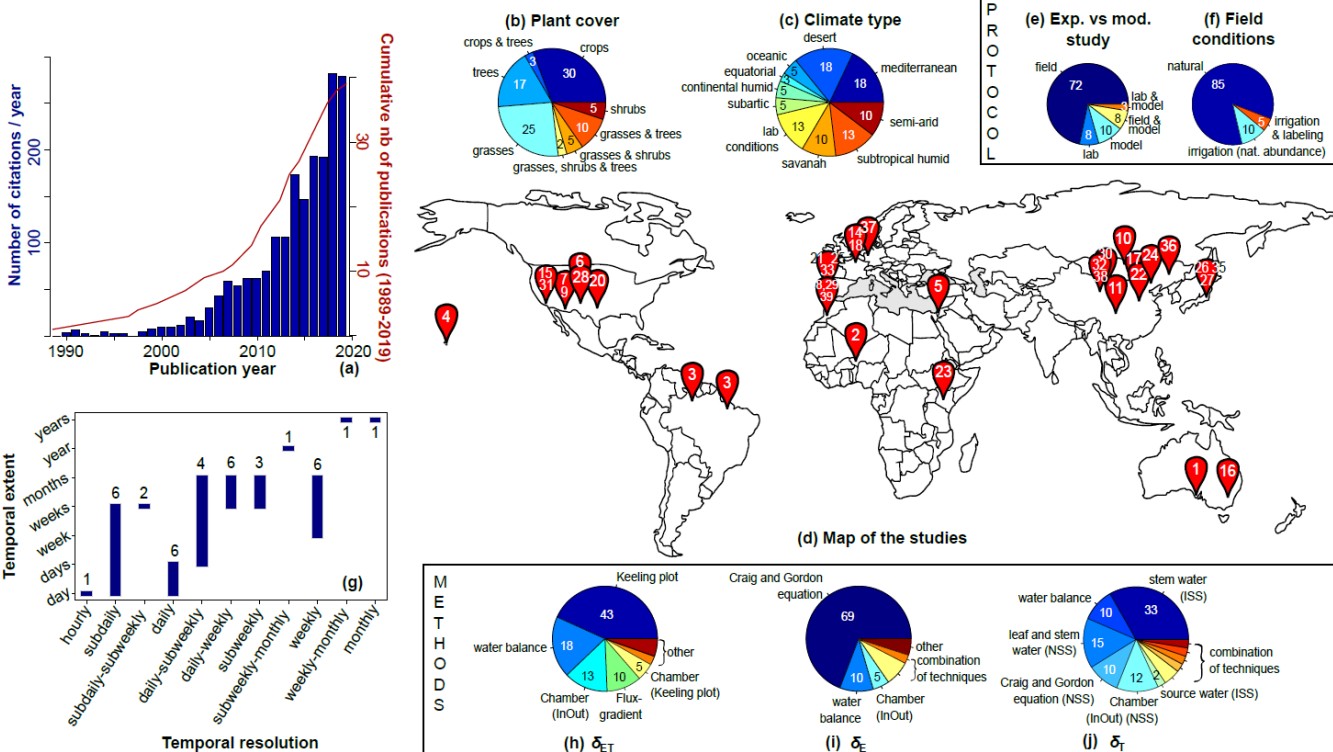

**Figure 1: Graphical summary of the reviewed literature. (a) Evolution of the number of citations per year (blue bars) and cumulative number of publications (1989-2020, red line); (b) Temporal resolution vs. extent of the estimate of transpiration to evapotranspiration ratio (T/ET). Numbers above/below the histograms refer to the number of studies working at a given temporal resolution; (c) and (d) Listing of the different plant cover and climate types with proportions (white label) expressed in percentage and (g) map locating each study (with reference number #1-39 in white label, see Table A2); (e) and (f) Proportions of field vs. modelling studies and prevailing experimental conditions (natural precipitation or irrigation, or else labelling studies); (h)-(j): listing and proportions of methods for determination of $\delta_{ET}$, $\delta_E$, and $\delta_T$.**





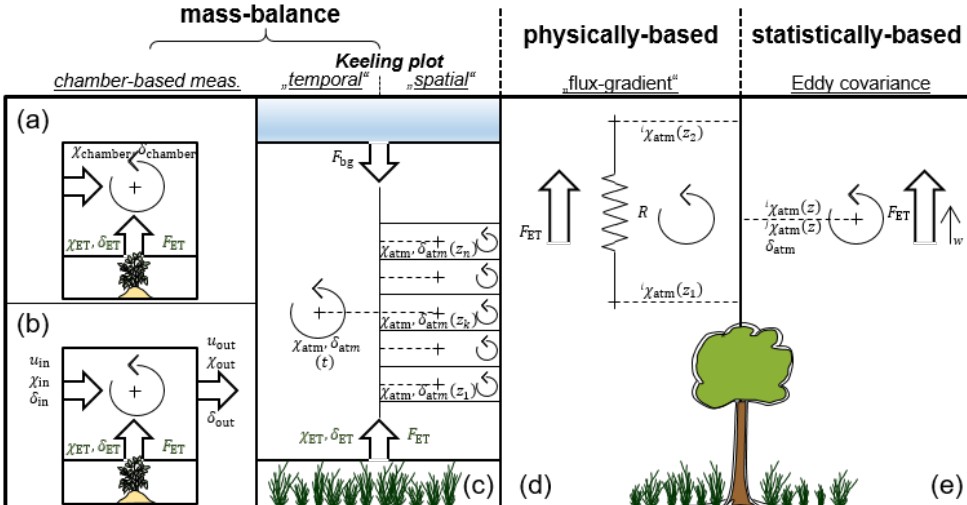


**Figure 2. Summary of the different approaches (mass-balance, physically- and statistically-based) methods for determination of $\delta_{ET}$ with the relevant variables and fluxes for each case.**



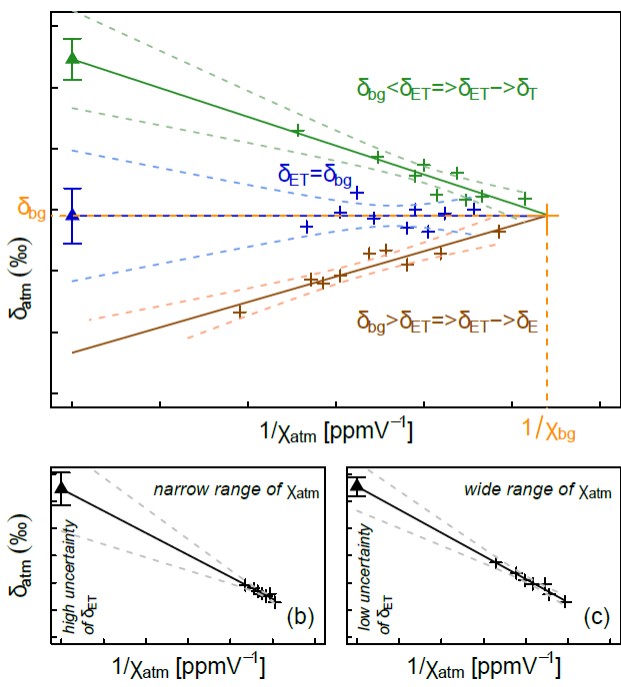

**Figure 3. Illustration of the Keeling (1958) plot technique for determination of the isotopic composition of the surface flux, here evapotranspiration ($\delta_{ET}$). Subscript "bg" refers to the atmospheric background air, i.e., the air, which is not influenced by the surface ET flux. (a) Cases with different slopes of the regression line and implications for the nature of the surface flux: ET tends either toward transpiration (T) or evaporation (E). Illustration of the importance of the (narrow or wide) spread in water vapour mixing ratio ($\chi_{atm}$, ppmV) values for the uncertainty of the $\delta_{ET}$ estimate (panels b and c).**



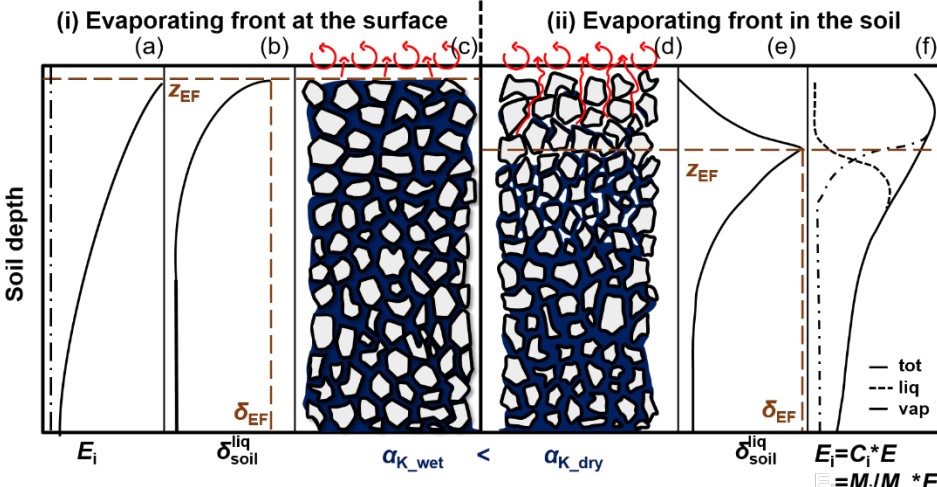

**Figure 4.** Effect of the water status of the soil, i.e., the positioning of the evaporating front (EF, dashed line), on the value of kinetic fractionation factor ($\alpha_K$). Panels a-c refer to the situation of a saturated soil (subscript "wet") where the EF is located at the soil surface; panels d-f refer to a dry soil with the EF below the soil surface. The corresponding soil water total (solid line), liquid (dotted line), and vapour (dash-dotted line) isotopic flux profiles ($E_i$, [M L$^{-3}$]) (panels a/f), soil liquid isotopic composition profile (panels b/e) are reported as well. Adapted from Braud et al. (2005).





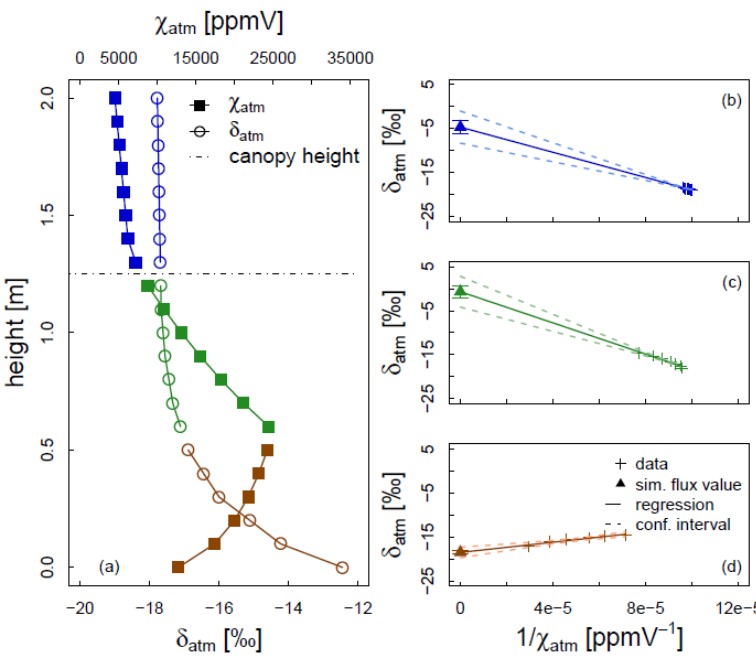


**Figure 5. One example of application of the Keeling (1958) plot technique to synthetic data that would be collected with a field-deployable lift at high vertical resolution (0.1 m) (for implications on measurement frequency, which also needs to be high (≥5Hz), see sections 3.1 and 4 of Ney and Graf (2018)). The oxygen isotopic compositions of evapotranspiration, transpiration, and evaporation are estimated by the values of the y-intercepts of the linear regressions between the isotopic composition of the**
**atmospheric water vapour ($\delta_{atm}$) and the inverse of the water vapour mixing ratio ($\chi_{atm}$) in three non-overlapping regions, i.e., (i) the "free atmosphere" (indicated by the blue symbols), (ii) the region spreading from the canopy height to the height of local maximum in $\delta_{atm}$ (green symbols), and (iii) the region delimited by the $\delta_{atm}$ local maximum height and the ground level (brown symbols). Also shown: 95 % confidence interval envelopes of the linear regressions (dashed lines) as well as error bars (1 standard error) of the y-intercepts.**






## 8 Appendix

**Table A1: List of Symbols and Abbreviations used in the main document and Table A2**

| Symbol or abbreviation | Description | Dimension or unit |
|---|---|---|
| $C_{atm}$, $C_{bg}$, $C_{ET}$ | Atmospheric and background water vapour concentration, rise in atmospheric water vapour concentration due to evapotranspiration flux | $M\,L^{-3}$ |
| $E$, $E_{Lys}$, $E_{pot}$ | Soil evaporation rate, soil evaporation (micro-lysimeter), potential evaporation | $L^3\,T^{-1}$ |
| EC | Eddy Covariance | |
| ET | Evapotranspiration rate | $L^3\,T^{-1}$ |
| $f$ | Measurement frequency | $T^{-1}$ |
| $F_{ET}$, $^iF_{ET}$, $^iF_E$, $F_E$, $^jF_E$, $^jF_T$, $^iF_T$, $^jF_{xyl}$, $^iF_{xyl}$ | Evapotranspiration water vapour flux density rate, Evapotranspiration, evaporation, transpiration, and xylem water flux density rates of the rare (i) and abundant (j) isotopologue | $L^3\,L^{-2}\,T^{-1}$ |
| $g_s$ | Leaf stomatal conductance | mmol m$^{-2}$ s$^{-1}$ |
| GPP | Gross primary production | $M\,L^{-2}\,T^{-1}$ |
| $h_{atm}$, $h_{EF}$ | Atmospheric relative humidiy and soil pore space relative humidity at the evaporating front | - |
| $K$ | Eddy diffusivity of water vapour | $L^2\,T^{-1}$ |
| LAI | Leaf area index | $L^2\,L^{-2}$ |
| $L_{ET}$ | latent heat flux of evapotranspiration | $M\,T^{-3}$ |
| $M_{atm}$, $M_w$ | Dry air and water molecular weight | $M\,L^{-3}$ |
| $n$ | Adimensional factor accounting for flow conditions above the liquid water-water vapour equilibrium layer | - |
| NEE | Net ecosystem exchange | $M\,L^{-2}\,T^{-1}$ |
| $P$ | Precipitation amount | $L^3\,L^{-2}$ |
| $p$ | Proportion of leaf water in isotopic equilibrium with water vapour in the stomatal cavity | - |
| PPFD | Photosynthetic photon flux density | µmol s$^{-1}$ m$^{-2}$ |
| $Q_s$ | Sensible heat flux | $M\,T^{-3}$ |
| $R$ | Universal gas constant | $M\,L^{-1}\,T^{-3}$ |
| $R_{std}$, $R_{EF}$, $R_{sat}$ | Isotopic ratio of the Vienna Standard Mean Ocean Water (V-SMOW), soil water at the evaporating front, and of saturated water vapor | - |
| $R_n$, $R_s$, $R_d$ | Net and solar radiation, and radiation flux density | $M\,T^{-3}$ |
| $R_{gas}$ | Universal gas constant | $M\,L^{-1}\,T^{-3}$ |
| $^ir$, $^ir$ $^jD$, $^iD$ | Bulk resistances to vapour transport of the rare (i) and abundant (j) isotopologues Molecular diffusivities of the rare (i) and abundant (j) water vapour isotopologues | |
| $S$ | Sap-flux density | $M\,L^{-2}\,S^{-1}$ |
| $T$ | Transpiration rate | $L^3\,T^{-1}$ |
| $T_{atm}$, $T_{soil}$, $T_{EF}$, $T_L$, $T_{can}$ | Temperature of the atmosphere, soil, soil at the evaporating front, leaf surface, and canopy atmosphere | °C |
| T/ET | Transpiration fraction | - |
| $u_{in}$, $u_{out}$ | Flow rate measured at the inlet and outlet of a gas exchange chamber | $L^3\,T^{-1}$ |
| $v$ ($v_d$) | Wind speed (wind direction) | $L\,T^{-1}$ |
| VFC | Vegetation fractional coverage | $L^2\,L^{-2}$ |
| VPD | Vapour pressure deficit | P |
| $z$, $z_{atm}$, $z_{EF}$ | Vertical coordinate, atmospheric height, and soil evaporating front depth | M |
| $\alpha_{eq}$ $\alpha_K$ | Equilibrium isotopic fractionation factor Kinetic isotopic fractionation factor | - |





| $\delta_{atm}$, $\delta_{soil}$, $\delta_{soil}^{vap}$ $\delta_L$, $\delta_{xyl}$, $\delta_{prec}$, $\delta_{root}$, $\delta_{irr}$, $\delta_{pond}$, $\delta_{in}$, $\delta_{out}$ | Isotopic composition of the atmospheric water vapour, soil water, soil water vapour, leaf water, xylem water, precipitation, root water, irrigation water, pond water, water vapour measured at the inlet and outlet of a gas exchange chamber | - |
|---|---|---|
| $\varepsilon_{eq}$ | Equilibrium isotopic fractionation | - |
| $\varepsilon_K$ | Kinetic isotopic fractionation | |
| $\varphi$ | Isotope analyser inlet flow rate | $L^3\,T^{-1}$ |
| $\rho_{atm}$, $\rho_w$ | Dry air and water volumetric mass | $M\,L^{-3}$ |
| $\theta_{soil}$, $\theta_{surf}$, $\theta_{res}$, $\theta_{sat}$, $\theta_L$ | Soil, soil surface, soil residual, and soil saturated water content, Leaf water content | $L^3\,L^{-3}$ |
| $\tau_L$ | Leaf water turnover time | $T$ |
| $\chi_{atm}$, $\chi_{bg}$, $^j\chi_{atm}$, $^i\chi_{atm}$, $^j\chi_{atm}^{sat}$, $^i\chi_{atm}^{sat}$, $\chi_{ET}$, $\chi_{in}$, $\chi_{out}$ | Atmospheric and background water vapour mixing ratio, water vapour mixing ratio in rare (i) and abundant (j) isotope, saturated water vapour mixing ratio in rare (i) and abundant (j) isotope, rise in atmospheric water vapour mixing ratio due to evapotranspiration flux, water vapour mixing ratio measured at the inlet and outlet of a gas exchange chamber | $L^3\,L^{-3}$ |
| $\psi_{EF}$ | Soil water matric potential at the evaporating front | $M\,L^{-1}\,T^{-2}$ |






**Table A2: Overview of the partitioning studies found with the ISI Web of Science search engine (http://www.webofknowledge.com/) on basis of search terms (("evapotranspiration" or "transpiration" or "evaporation") and partition\* and isotop\*). 'CG65', 'Kp58', and 'f-g' refer respectively to the Craig and Gordon (1965) equation for determination of $\delta_E$ and $\delta_T$, the Keeling (1958) plot and flux-gradient techniques for determination of $\delta_{ET}$ and $\delta_E$. 'Chamber(InOut)' and 'Chamber(Kp58)' refer to gas exchange chamber-based measurements for determination of $\delta_{ET}$, $\delta_E$, and $\delta_T$ by either comparing the chamber inlet and outlet gas properties or by applying the Keeling plot technique, respectively. The reader is referred to Table A1 for the definitions of the other symbols and abbreviations.**

| Nb | Author (Year) | Field, Lab, or Model | Location | Land surface type (LAI / VFC) | Climate, $T_{atm}$, $P$ | Isotopic measurements with range of meas. heights (m) / depths (cm) and sampling intervals | additional measurements | Temporal resolution (extent) | Extraction technique | Measurement technique | | | T/ET results |
|----|----|----|----|----|----|----|----|----|----|----|----|----|----|
| | | | | | | | | | | $\delta_{ET}$ | $\delta_E$ | $\delta_T$ | |
| 1 | Walker and Brunel (1990) | Field | Hincks Conservation Park, Australia | Eucalyptus mallee (69 %) | Semi-arid, 30°C (Jan), 23.6 °C (Mar) 400 mm (annual) | $\delta_{atm}$ (2.25-9,2.25<int<4.50), $\delta_{soil}$ (0-200,10<int<20), $\delta_{xyl}$, $\delta_L$ | $H$, $T$, $T_{atm}$, $T_{soil}$, $T_L$ | daily (days) | Azeotropic distillation | Isotope mass-balance | CG65 | CG65 (NSS) and leaf and stem water (NSS) | T has the largest contribution to ET |
| 2 | Brunel et al. (1997) | Field | Sahel, Niger | Fallow bushland of woody shrubs (20 %) | Semi-arid 65 mm (exp. period) | $\delta_{atm}$ (3-12,3<int<6), $\delta_{soil}$ (0-120; int=10), $\delta_{xyl}$, $\delta_{prec}$ | $F$, $h$, $T_{atm}$, $T_{soil}$, $\theta_{soil}$ | weekly (weeks) | Azeotropic distillation | Isotope mass-balance | CG65 | Stem water (ISS) | 21 % |
| 3 | Moreira et al. (1997) | Field | Amazon basin | Indigenous forest (5-6.1) | Tropical 1750-2000 mm | $\delta_{atm}$ (0-45, int~20), $\delta_{soil}$ (0), $\delta_{xyl}$ | $F$, $h$, $T_{atm}$ | daily (day) | Direct equilibration with $CO_2$ | Kp58 | CG65 | Stem water (ISS) | T potentially a major source of water vapour during the dry season |
| 4 | Hsieh et al. (1998) | Field | Hawaii | Not reported | Savanah 17-23°C 180-2500 mm | $\delta_{soil}$ (0-70, 5<int<20), $\delta_{prec}$ | $E_{pot}$ | weekly-monthly (years) | Direct equilibration with $CO_2$ | Isotope mass-balance | | | 14-71 % |
| 5 | Wang and Yakir (2000) | Field | Negev region, Israel | Wheat field | Desert | $\delta_{atm}$ (0.8-70,1<int<26), $\delta_{soil}$, $\delta_{xyl}$, $\delta_L$ | $h$, $T_{atm}$, $T_{soil}$ | daily-weekly (months) | Cryogenic vacuum distillation | Kp58 | CG65 | Stem water (ISS) | 96.5-98.5 % during midday |
| 6 | Ferretti et al. (2003) | Field | Colorado, USA | Shortgrass steppe | Semi-arid, 15.6 °C (summer), 0.6 °C (winter), 320 mm (annual) | $\delta_{soil}$ (1-50,1<int<25), $\delta_{prec}$ | Bowen Ratio ($Q_g/L_{ET}$), $R_n$, $T_{atm}$, $T_{soil}$, $\theta_{soil}$ | monthly (years) | Direct equilibration with $CO_2$ | Isotope mixing model | | | 10-60 % |
| 7 | Yepez et al. (2003) | Field | Arizona, USA | Savanna woodland (1.6) *Sporobolus wrightii* (grass), Prosopis velutina (trees) | Savanah 24.8 °C (Jul), 9.9 °C (Jan) 343 mm (annual) | $\delta_{atm}$ (0.1-14,0.4<int<2), $\delta_{soil}$ (0-10), $\delta_{xyl}$ | $h$, LAI, $R_s$, $T_{atm}$, $T_{soil}$, $v$ | None | Cryogenic vacuum distillation | Kp58 | CG65 | Stem water (ISS) | 70 % (Tree) 15 % (grass) |
| 8 | Williams et al. (2004) | Field | Marrakech, Morocco | Olive orchard | Mediterranean 253 mm | $\delta_{atm}$ (0.1-8.9,1<int<25), $\delta_{soil}$ (1-25,int=25), $\delta_{xyl}$ | $h$, $L_{ET}$, $Q_s$, $R_s$, $T_{atm}$, $T_{soil}$, $v$, $v_d$ | subdaily (days) | Cryogenic vacuum distillation | Kp58 | CG65 | Stem water (ISS) | Prior irrigation: 100 % following irrigation: 69-85 % |
| 9 | Yepez et al. (2005) | Field | Arizona, USA | Grassland (*E.lehman-niana*, 0.66; *H.contortus*, 0.37) | Semi-arid (savanah) 39 mm (irrigation pulse) | $\delta_{soil}$ (1-25, 2<int<10), $\delta_L$ | $e$, $g_s$, $h$, LAI, $T_{atm}$, $T_{can}$ | subdaily (week) | Cryogenic vacuum distillation | Chamber (Kp58) | CG65 | Leaf and stem water (NSS) | Prior irrigation: 35(±7) %, after irrigation: 22(±5)-43(±8) % |
| 10 | Tsujimura et al. (2007) | Field | Easter Mongolia | Grassland (*Stipa krylovii*, *Carex spp.*, and *Artemisia spp.*, 0.21-0.57) | Semi-arid (subartic) 150-300 mm | $\delta_{atm}$ (0.5-10, 25<int<500), $\delta_{soil}$ (50-150), $\delta_{prec}$ | $L_{ET}$, $h$, $P$, $T_{atm}$, $\theta_{soil}$ | daily-subweekly (days) | Cold distillation | Kp58 | CG65 | Source (soil) water (ISS) | 60-73 % (forest site) 35-59 % (grassland site) |
| 11 | Xu et al. (2008) | Field | Balang Mountain, China | Subalpine shrubland (2.05) | Oceanic 3°C (annual) 710 mm (annual) | $\delta_{atm}$ (0.1-3,0.4<int<1), $\delta_{soil}$ (0-10), $\delta_{xyl}$ | $E_{pot}$, $h$, LAI, $P$, $v$, PPFD, $T_{atm}$, $T_{soil}$ | daily (days) | Cryogenic vacuum distillation | Kp58 | CG65 | Stem water (ISS) | 65.6(±8.3)-96.9(±2.0) % |
| 12 | Wenninger et al. (2010) | Lab | Delft, Netherlands | Bare soil and Teff crop | Lab. conditions | $\delta_{soil}$ (1.7-22,3.4<int<7.5) | EC, $T_{soil}$, $\theta_{soil}$ | subweekly (weeks) | na (soil moisture sensors) | Isotope mass-balance | | | 70 % |
| 13 | Wang et al. (2010) | Lab | Arizona, USA | Mesquite tree (25-100 %) | Lab. conditions | $\delta_{atm}$ (0.5-2,0.5<int<1), $\delta_{irr}$ | $h$, $T_{atm}$, $T_{soil}$ | hourly (day) | na | Kp58 | CG65 | Chamber (InOut,Kp58) | 61-83 % |
| 14 | Rothfuss et al. (2010) | Lab | Lab. conditions | Tall fescue cover (0-3.9) | Lab. conditions | $\delta_{atm}$, $\delta_{soil}$ (0-12,int~1), $\delta_{xyl}$ | $h$, LAI, $T_{atm}$, $\theta_{soil}$ | weekly (weeks) | Cryogenic vacuum distillation | Condensed water | Groundwater (ISS) | Stem water (ISS) | 0-95 % |
| 15 | Bijoor et al. (2011) | Field | Orange County, USA | Freshwater marsh *typha latifolia* | Mediterranean 16.4 °C (annual) 270 mm (annual) | $\delta_{atm}$ (0.1 and 4.0), $\delta_{soil}$ (0-5), $\delta_L$, $\delta_{root}$ | EC, $h$, $L_{ET}$, $T_{atm}$, $v$ | subweekly-monthly (year) | Cryogenic vacuum distillation | Isotope mass-balance | CG65 | Chamber (InOut, NSS) – Root water (ISS) | 56-67 % |
| 16 | Haverd et al. (2011) | Field & model | Southern Australia | Eucalyptus forest | Temperate | $\delta_{atm}$ (2.0, 4.4, 10.4, 26.3, 35.4, 43.4, 70.1m) | $L_{ET}$, $Q_s$, $T_{atm}$, $v$, $\theta_L$ | subdaily (weeks) | Cryogenic vacuum distillation (plant) | SVAT model | chamber (InOut) | Chamber (InOut, NSS) | 85(±2) % |
| 17 | Zhang et al. (2011) | Field | North China Plain, China | Irrigated winter wheat (2.6) | Subtropical humid 12 °C (annual), 480 mm (annual) | $\delta_{atm}$ (0.1, 3, 10), $\delta_{soil}$ (20-100, 10<int<20), $\delta_{xyl}$, $\delta_{prec}$ | $h$, LAI, $L_{ET}$, $T_{atm}$, $\theta_{soil}$ | weekly (week) | Cryogenic vacuum distillation | Kp58 | CG65 | Stem water (ISS) | 60-83 % |
| 18 | Rothfuss et al. (2012) | Lab & model | Lab. conditions | Tall fescue cover (0-3.9) | Lab. conditions | $\delta_{soil}$ (0-12), $\delta_{xyl}$ | $h$, LAI, $T_{atm}$, $\theta_{soil}$ | weekly (weeks) | Cryogenic vacuum distillation | Chamber condensed water vapor | Groundwater (ISS) | Stem water (ISS) | 0-95 % |
| 19 | Sutanto et al. (2012) | Lab & Model | Delft, Netherlands | Grass-covered lysimeter | Lab. conditions | $\delta_{soil}$ (7-33, int=7) | EC, $h$, $R_s$, $T_{atm}$, $T_{soil}$, $v$, $\theta_{soil}$ | subweekly (months) | na (soil moisture sensors) | Isotope mixing model | | | 87% (HYDRUS 1D: 70%) |
| 20 | Wang et al. (2013) | Field | Oklahoma, USA | Grassland | Subtropical humid 16 °C (annual), 911 mm (annual) | $\delta_{soil}$ (0-2) | $h$, $T_{atm}$, $T_{soil}$, $\theta_{soil}$ | None | Cryogenic vacuum distillation | Chamber (Kp58) | 1. bare soil chamber (Kp58 and InOut) 2. CG65 | Chamber (Kp58 & InOut, NSS) – stem water (ISS) | 65-86 % |
| 21 | Dubbert et al. (2013) | Field | Central Portugal | Open cork-oak woodland | Mediterranean 15.9 °C (annual) 680 mm (annual) | $\delta_{soil}$ (0.5-40, 3<int<20) | $h$, $P$, PPFD, $T_{atm}$, $T_{soil}$, $\theta_{soil}$ | subdaily-subweekly (week) | Cryogenic vacuum distillation | Chamber (InOut) | 1. CG65 2. chamber (InOut) | Leaf and stem water (NSS) | 50-80 % |
| 22 | Sun et al. (2014) | Field | Yellow River Xiaolangdi forest, China | Chinese cork oak (96 % vegetation) | Mediterranean 13.4 °C (annual) 643 mm (annual) | $\delta_{atm}$ (0.1, 11, 18), $\delta_{soil}$ (2.5-7.5,int=5), $\delta_{xyl}$, $\delta_L$ | $g_s$, $h$, $T_{atm}$, $T_L$, $v$, $v_d$, $\theta_{soil}$ | subdaily (day) | Cryogenic vacuum distillation | Kp58 | CG65 | CG65 (NSS) | 85-91 % |
| 23 | Good et al. (2014) | Field | Mpala Research Center, Kenya | Grassland (0-10%) | Semi-arid (savanah) 30 mm (irrigation), 6.7 mm (rain) | $\delta_{atm}$ (0.4), $\delta_{soil}$ (1-20, 5<int<10), $\delta_L$, $\delta_{prec}$, $\delta_{irr}$ | LAI, $L_{ET}$, $Q_s$, $R_n$, $T_{atm}$, $T_{soil}$, $\theta_{soil}$ | daily (days) | Cryogenic vacuum distillation | Kp58 | CG65 | Chamber (InOut, NSS) | 29(±5) % (mean value) 40 % (max. value) |





| # | Reference | Type | Location | Vegetation | Climate | δ notation | Variables | Temporal resolution | Water extraction | T model | E method | Source | T/ET |
|---|---|---|---|---|---|---|---|---|---|---|---|---|---|
| 24 | Hu et al. (2014) | Field | Mongolia, China | Grassland (0.4-0.55) | Semi-arid 2.1 °C (annual), 18.9 °C (Jul), −17.5°C (Jan) 383 mm (annual) | $\delta_{atm}$ (0.7, 1.7), $\delta_{soil}$ (5-25, int=10), $\delta_{xyl}$, $\delta_L$ | LAI, $L_{ET}$, $T_{atm}$, $T_{soil}$, $T_{can}$, $\theta_{soil}$ | subdaily-subweekly (weeks) | Cryogenic vacuum distillation | f-g | CG65 | CG65 (NSS) | 83 % |
| 25 | Dubbert et al. (2014b) | Field | Central Portugal | Open cork-oak woodland (1.05) | Mediterranean, 15.9 °C (annual) 680 mm (annual) | $\delta_{atm}$ (2), $\delta_{soil}$ (0-40, 2<int<20), $\delta_{prec}$ | h, LAI, $L_{ET}$, NEE, PPFD, P, $T_{atm}$, $T_{soil}$, $\theta_{soil}$, | daily-weekly (months) | Cryogenic vacuum distillation | Chamber (InOut) | CG65 | Leaf and stem water (NSS) | 45-84 % |
| 26 | Wei et al. (2015) | Field | Tsukuba, Japan | rice paddy field (0-5.5) | Subtropical humid 13.7 °C (annual) 1200 mm (annual) | $\delta_{atm}$ (2) | h, $T_{atm}$, LAI, $L_{ET}$ | daily-weekly (months) | na | Kp58 | CG65 | Source (pone) water (ISS) | 2 – 100 % |
| 27 | Wang et al. (2015) | Field & Model | Tsukuba, Japan | Grassland (0.01-2.58) | Subtropical humid 14.1 °C (annual) 1159 mm (annual) | $\delta_{atm}$ (0.1-2, 0.4<int<1), $\delta_{soil}$ (5 depths), $\delta_{xyl}$, $\delta_L$ | $g_s$, h, $L_{ET}$, LAI, $Q_s$, $R_n$, $R_s$, $T_{atm}$, $T_{soil}$, $T_L$, $\theta_{soil}$, $\theta_L$ | weekly (months) | Cryogenic vacuum distillation | Kp58 | CG65 | Leaf and stem water (NSS) | 2-99 % |
| 28 | Berkelhammer et al. (2016) | Field & Model | Rocky Mountains National Park, USA | Subalpine coniferous forest (1.2-4.2) | Site1: 14 °C (July), 884 mm (annual) Site2: 19 °C (July) 430 mm (annual) | Site1: $\delta_{atm}$ (10-20,int=5); Site2: $\delta_{atm}$ (12-25.1,5.7<int<8.4), $\delta_{xyl}$, $\delta_L$ | GPP, LAI, $L_{ET}$, $Q_s$, $T_{atm}$, VPD, $\theta_{soil}$ | weekly (months) | na | modified Kp58 (Noone et al., 2013) | CG65 | Leaf and stem water (NSS) | 49(±23) % |
| 29 | Aouade et al. (2016) | Field | Haouz plain, Marocco | irrigated winter wheat (0-1.2) | Semi-arid 240 mm (annual) | $\delta_{atm}$ (0-3,1<int<1.6), $\delta_{soil}$ (0-70, 2<int<10), $\delta_{xyl}$ | h, p, P, $R_s$, $T_{atm}$, $T_{soil}$, v, $\theta_{soil}$ | daily (days) | Cryogenic vacuum distillation | Kp58 | CG65 | Stem water (ISS) | 73-89 % |
| 30 | Wen et al. (2016) | Field | Heihe River Basin, China | spring maize, (5.6) | Semi-arid, 74. °C (annual), 129.7 mm (annual) | $\delta_{atm}$ (0.5,1.5), $\delta_{soil}$ (2.5-80, 5<int<10), $\delta_{xyl}$, $\delta_L$, $\delta_{prec}$, $\delta_{irr}$ | h, LAI, $L_{ET}$, P, $T_{atm}$, $T_{soil}$, v, $\theta_{soil}$ | daily-weekly (months) | Cryogenic vacuum distillation | f-g | CG65 | CG65 (NSS) | 87(±5) % |
| 31 | Lu et al. (2017) | Field | California, USA | Desert Valley: forage sorghum (0.5-1.5) | arid 22.4 °C (annual), 12.6 °C (Jan), 32.9 (Aug) 80.3 mm (annual) | $\delta_{atm}$ | h, LAI, $L_{ET}$, P, $R_s$, $T_{atm}$, $T_{soil}$, v | daily-subweekly (days) | na | Chamber (InOut) | Chamber (InOut) | Chamber (InOut, NSS) | 46(±6) % |
| 32 | Wu et al. (2017) | Field | Gansu Province, China | University test field: maize (0-4) | arid, 8 °C (annual) 164 mm (annual) | $\delta_{atm}$ (1,2,4), $\delta_{soil}$ (2.5,7.5), $\delta_{xyl}$ | h, $L_{ET}$, $T_{atm}$, $T_{soil}$, v | subweekly (months) | Cryogenic vacuum distillation | Chamber (InOut) | 1. Chamber (InOut) 2. CG65 | Chamber (InOut, NSS) | 59-87 % |
| 33 | Piayda et al. (2017) | Field | Central Portugal | Open cork-oak woodland: oak and grass (1.1) | Mediterranean 15.9 °C (annual) 680 mm (annual) | $\delta_{soil}$ (0-40, 2<int<20) | LAI, P, PPFD, $T_{atm}$, $T_{soil}$, $\theta_{soil}$ | daily-subweekly (days) | Cryogenic vacuum distillation | Chamber (InOut) | CG65 | CG65 (NSS) | 9-59 % (open) 17-66 % (shaded) |
| 34 | Wei et al. (2018) | Field & Model | Japan and China | Rice field (0-6), winter wheat and summer corn (0-4.7) | 13.7 °C (annual) 1200 mm (annual) | $\delta_{atm}$ (2), $\delta_{soil}$ (2.5-45, 15<int<25), $\delta_{xyl}$, $\delta_L$ | h, LAI, $L_{ET}$, P, $Q_s$, $R_d$, $R_n$, $T_{atm}$, $T_{soil}$, v, $v_d$ | daily-subweekly (months) | Cryogenic vacuum distillation | 1: Kp58 2: f-g | CG65 | Stem water (ISS) | 74 % (rice), 93 % (wheat), 81% (corn) |
| 35 | Zhou et al. (2018) | Field | Heihe River Basin (HRB), China | Alpine meadow (6.3), irrigated maize (3.8), and Populus euphratica (0.8) | upper HRB: −0.4 °C (annual) 438 mm (annual) middle HRB: 6.9 °C (annual) 147 mm lower HRB: 10.4 °C (annual) 26 mm (annual) | $\delta_{atm}$ (0.5,1.5), $\delta_{soil}$ (2.5-80, 5<int<10), $\delta_{xyl}$, $\delta_L$ | F, LAI, $L_{ET}$, NEE, P, $Q_s$, r, $R_n$, $T_{atm}$, $T_{soil}$, v, $v_d$, $\theta_{soil}$ | daily-weekly (months) | Cryogenic vacuum distillation | f-g | CG65 | Stem water (ISS) – leaf and stem water (NSS) | 72-100 % |
| 36 | Zhang et al. (2018) | Field | Jilin Province, China | S. triqueter (0.16) and P. australis (0.86) | Semi-arid 4.2 °C 392 mm | $\delta_{atm}$ (0.2,0.9,1.9 cm), $\delta_{xyl}$, $\delta_{prec}$, $\delta_{pond}$ | h, LAI, $T_{atm}$, $\theta_L$ | subdaily (days) | Cryogenic vacuum distillation | Not used | CG65 | Leaf and stem water (NSS) | 20% (S. triqueter) 20% (P australis) |
| 37 | Quade et al. (2019) | Field | Selhausen, Germany | Sugar beet | Oceanic 18.6 °C (exp. period) 207.8 mm (exp. period) | $\delta_{atm}$ (0.01-1.50,0.19<int<0.5), $\delta_{soil}$ (1-10,int=5), $\delta_L$ | h, LAI, $L_{ET}$, $T_{atm}$, $T_{soil}$ | subdaily (days) | Cryogenic vacuum distillation (plant) / non destructive monitoring (soil) | Kp58 | CG65 | Stem water_(ISS) | 57-74 % |
| 38 | Xiong et al. (2019) | Field | Heihe River Basin (HRB), China | spring maize, (5.6) | Desert 7.3 °C 100-250 mm | *See Wen et al. (2016)* | | daily-weekly (weeks) | Cryogenic vacuum distillation | f-g | CG65 | Stem water_(ISS) - leaf and stem water (NSS) | 54-97% 85 % (mean value) |
| 39 | Aouade et al. (2020) | Model | Haouz plain, Marocco | irrigated winter wheat (0-1.2) | Semi-arid 240 mm (annual) | *See Aouade et al. (2016)* | | daily (days) | Cryogenic vacuum distillation | Kp58 | CG65 | Stem water_(ISS) | 80 % |



**Acknowledgements**

Youri Rothfuss and Maren Dubbert acknowledge funding by the German Science Foundation DFG (grant numbers RO 5421/1-1 and #DU1688/1-1). Maria Quade was funded by the German Ministry of Education and Research BMBF within the project
IDAS-GHG (Instrumental and Data-driven Approaches to Source Partitioning of Greenhouse Gas Fluxes: Comparison, Combination, Advancement, grant number 01LN1313A).

**Competing interests**

The authors declare that they have no conflict of interest.

**Author contribution**

Youri Rothfuss, Maria Quade, and Maren Dubbert reviewed the published literature and prepared the manuscript. All authors reviewed the manuscript.

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
