# Peer review of "Reviews and syntheses: Gaining insights into evapotranspiration partitioning with novel isotopic monitoring methods"

_Biogeosciences, 2020_

## Referee Comment (RC1) · John Marshall (Referee) · 18 Dec 2020

This paper addresses the use of isotopic data to partition evaporative water fluxes from terrestrial ecosystems, specifically focusing on transpiration vs. soil evaporation. It notes that isotopic methods have frequently been labelled "powerful," but have in fact proven difficult to use. The authors describe the barriers and complexities and propose several preferred pathways forward. It seems that these methods may be ready, at long last, to leap over the barriers that have constrained them. This paper prepares us for that leap by focusing attention on emerging best practices.

The topic is deeply relevant to BG because water fluxes control so many biogeochemical processes. This is so because, first, transpiration and soil evaporation are controlled by different environmental variables, but also because the water is drawn from different depths in the soil, which results in different vertical patterns of soil moisture. Third, the resulting moisture profiles affect many biogeochemical processes and fourth, the water fluxes redistribute solutes. It seems that these methods may be ready, at long last, to leap over the barriers that have constrained them. This paper prepares us for that leap by focusing attention on emerging best practices.

The paper begins with a review of earlier work and finishes with suggestions about how to move forward. Its novelty lies in the detection and presentation of trends in the broader literature and in the identification of key results in recent papers. These key results are mostly methodological. Because of this structure, the paper does not so much come to novel conclusions as emphasize promising methods.

It might be helpful to begin with a summary of where the large variation in T/ET comes from. Before the series of T/ET values in Section 2, it would be useful to note that some of this variation is probably real and that some of the variation is predictable, e.g., when soils range from wet to dry or canopies range from isolated seedlings to closely spaced mature plants. The range of T/ET estimates in Section 2 would then make more sense. Also, I understand that Section 2 is intended as a timeline, but I wonder if it could be provided a bit more narrative flow. In particular, topic sentences at the beginnings of the paragraphs would help, if this is possible.

I was puzzled by the fact the isotopic estimates of T/ET were not directly compared to other methods. This is especially surprising because (Sutanto et al., 2014), which is not cited, made this comparion and concluded that the isotopic data yielded lower estimates. It would seem that this discrepancy should be presented and discussed. This could also precede Section 2.

An important strength of Section 3 lies in the text descriptions of the meanings and assumptions of the mathematical models that have been used in this literature. This is

not an easy sort of writing, but these authors do it well. My only general criticism was that I found it difficult to determine which of these insights were drawn from previous work and which are new here.

In section 3.2, the authors discuss the isotopic signal of evaporation, focusing almost exclusively on soil evaporation while ignoring evaporation from plant surfaces (interception). Canopy evaporation can consume a substantial part of precipitation and it has isotopic consequences. There is some literature on the latter topic that includes descriptions of its isotopic consequences. The Allen et al. review (Allen et al., 2017) is one place to start. Much of the interception literature comes from forest canopies, but the same processes occur in crop canopies (e.g, Zheng et al., 2019). I presume that the models presented here include interception as part of transpiration, but perhaps they simply neglect wetted canopies. In either case, the treatment of interception should be explained clearly. It will be especially important in long-term estimates, in dense canopies, and where rainfall is frequent and light. I suppose it must also be much more important in sprinkler irrigation than in ditch or drip irrigation. If it is a research gap, I would highlight it in hopes that it will be addressed.

It is also critical to recognize the contribution of Braden-Behrens et al., (2019), who have applied eddy covariance techniques to water stable isotope data, as suggested by the authors. This would seem to fit around line 337, at the climax of the methods section.

I would also suggest more caution regarding the Keeling-Plot technique. Like the other methods described here, it is easily abused. As the authors note, the method depends on three important assumptions: first is that the method can only work if there are two—and only two– uniform water sources in the mixture. This can be problem along vertical canopy profiles, where the isotopic composition of evaporating water is likely to vary with the rooting depth of the different species and perhaps, with the humidity and isotopic composition of the atmosphere surrounding the leaves. Perhaps this problem is less severe in a crop monoculture than in mixed vegetation, but the issue should

be pointed out. The second problem with Keeling plots is that regression tends to flatten models fitted to noisy data, leading to incorrect estimates of the y-intercept. Because of this issue, some authors in the CO2 literature have recommended using the method only if the data meet fairly stringent requirements for R2. A wide range may help provide a high R2, but it does not guarantee it. Finally, there should be some discussion about which regression method should be used for the fitting of Keeling plots (Pataki et al., 2003; Wehr & Saleska, 2017).

Specific Comments: Line 16, 55: is it "powerful?" The manuscript argues otherwise later, both in a brief statement on lines 810-813 and in its overall tone. More than that, the Sutanto et al. (2014) review raises serious questions about this.

I would drop the first two paragraphs and replace them with the general description of T/ET requested above.

L67-70: the iss issue is important, but complicated, in part because it depends on one's objectives. I know these authors want to talk about this, but I would wait to raise it until later, when it can really be dealt with. L81:futuring is in sec 4, not 3, right? L85: progress L98: "noticeably low" and L104: "exceptionally high." Do the authors doubt these estimates? This should be clarified either as these comments are made or, perhaps better, in a final summary statement about true values of T/ET. As noted above, this paper would be strengthened by a general statement about what values T/ET should take and by a statement of how well the isotopic estimates match the alternatives. L150: the Péclet effect seems important here and it should be explained carefully. It is more than compartmentalization. The effect is described in a bit more detail on lines 640-641, but it is not named there. L265: explain ambient vs. backgd. Could this be called instead canopy vs. troposphere, for example? L280: deltaET, not ET L321: a plus or minus symbol missing? Also there is no earlier equation estimating C from a Keeling plot. If there were, you should cite it by number. L351-2: Not a strong diagnostic as the linear form can survive a linear change in either variable. 395: different footprint areas L460: hatm is determined by TDR? L582: A useful way to conclude this list of complications and worries would be to compare the estimated fluxes to empirical data from chambers or weighing lysimeters. This would allow the reader to decide how well these models work. I would do it with a figure. L719-720: this point is so important, but it is not clearly worded. I would say something like: "...assume ISS and hence treat $\delta$xyl as equal to $\delta$T. Although this assumption is probably justified for a daily integration, there is growing evidence that plants reach ISS only briefly in the course of a day, especially when environmental conditions change rapidly. Thus the analysis is greatly simplified by daily integration, if that is sufficient for the study objectives."

But perhaps the authors disagree?

L763: point, not punctual

References: Allen, S. T., Keim, R. F., Barnard, H. R., McDonnell, J. J., & Brooks, J. R. (2017). The role of stable isotopes in understanding rainfall interception processes: A review. WIREs. Water, 4(1), 1–17. https://doi.org/10.1002/wat2.1187 Braden-Behrens, J., Markwitz, C., & Knohl, A. (2019). Eddy covariance measurements of the dual-isotope composition of evapotranspiration. Agricultural and Forest Meteorology, 269–270, 203–219. https://doi.org/10.1016/j.agrformet.2019.01.035 Pataki, D. E., Ehleringer, J. R., Flanagan, L. B., Yakir, D., Bowling, D. R., Still, C. J., Buchmann, N., Kaplan, J. O., & Berry, J. A. (2003). The application and interpretation of Keeling plots in terrestrial carbon cycle research. Global Biogeochemical Cycles, 17(1). https://doi.org/10.1029/2001GB001850 Sutanto, S. J., van den Hurk, B., Dirmeyer, P. A., Seneviratne, S. I., Röckmann, T., Trenberth, K. E., Blyth, E. M., Wenninger, J., & Hoffmann, G. (2014). HESS Opinions "A perspective on isotope versus non-isotope approaches to determine the contribution of transpiration to total evaporation" Hydrology and Earth System Sciences, 18(8), 2815–2827. https://doi.org/10.5194/hess-18-2815-2014 Wehr, R., & Saleska, S. R. (2017). The long-solved problem of the best-fit straight line: Application to isotopic mixing lines. Biogeosciences, 14(1), 17–29. https://doi.org/10.5194/bg-14-17-2017 Zheng, J., Fan,

J., Zhang, F., Yan, S., Wu, Y., Lu, J., Guo, J., Cheng, M., & Pei, Y. (2019). Through-fall and stemflow heterogeneity under the maize canopy and its effect on soil water distribution at the row scale. The Science of the Total Environment, 660, 1367–1382. https://doi.org/10.1016/j.scitotenv.2019.01.104

———————————————————

---

## Referee Comment (RC2) · Anonymous Referee #2 · 20 Dec 2020

The submitted manuscript reviewed the methods of measuring or estimating $\delta$E, $\delta$T and $\delta$ET for identifying the possible challenges of these isotopic methods, and how they should progress in the future, especially novel non-destructive methods. While this study is meaningful for the use of isotopic partitioning methods, the current literature overview and methodological review did not offer enough supports for the forementioned objectives. However, before the paper can be published, a major revision for this manuscript will be needed. The general and specific comments are showed as follows: General comments: 1. Literature overview: A literature overview was listed in the section 2 as a timeline for highlighting the important progresses made over the past 30 years, however, it might be helpful to classify and summarize these literature. If pos-

sible, suggest to add the topic sentences at the beginnings of each paragraphs for understanding easily. 2. Methodological review: Suggest adding a theoretical schematic diagram including the flux of soil evaporation (E), transpiration (T), evapotranspiration (ET), and their stable isotopic compositions, and the key points in the estimation of $\delta$E, $\delta$T, and $\delta$ET in an ecosystem for understanding the calculation principle clearly. 3. Methodological review: One of assumptions of Keeling plot approach is temporal variations in the water vapor mixing ratio and $\delta$V are caused only by ET. However, rainfall, entrainment process of air, and so on can iniňĆuence the variation in $\delta$V at hourly or daily time scales, and introduces biases in the Keeling plot estimates. Suggest to add the analysis of the related uncertainties for Keeling plot approach. 4. Methodological review: Suggest to add the comprehensive comparisons of Keeling plot, flux gradient and EC isotopic iňĆux method according to the calculation principle. Keeling plot and flux gradient methods can be agreement under certain conditions, and both calculation principles of flux gradient and EC isotopic iňĆux methods is micrometeorological theory, however the discrepancy still exist among them (eg. Hu et al., 2020 etc.). Hu, Y., Xiao, W., Wei, Z., Welp, L., Wen, X., Lee, X. (2020) Determining the Isotopic Composition of Surface Water Vapor Flux From High-Frequency Observations Using Flux-Gradient and Keeling Plot Methods. Earth and Space Science, DOI: 10.1002/essoar.10501239.1. 5. Methodological review: How about the uncertainty analysis of EC isotopic iňĆux methods due to the loss of information during the covariance calculation between the isotopic compositions and vertical wind iňĆuctuations? 6. Methodological review: Suggest to clarifying the calculations and assumptions of converting ðİŻ£ value of water vapor into ðİŻ£ value of soil or xylem (liquid) water, and the analysis of possible uncertainty in detailed, for promoting the use of the novel non-destructive methods in the future. 3. Possible ways forward: It might be useful to focus on the improvement of difficult problems of recommended methods, for example, how the plant-size measurements based on the chamber methods are scaled up for canopy-level estimations. Specific comments: 1. Line 405: if the describe of "the second relies on destructive sampling of the soil and offline analysis of the extracted water" is suitable, and the nondestructive collection and online monitoring of the water vapour isotopic composition of soil atmosphere also can be used. 2. Line 640-670: Péclet effect is very important for these theoretical equations, however, its definition and quantification are not explicitly named here. 3. Line 735: For the cryogenic extraction methods of xylem water, new biases might be considered, because Chen et al. (2020) found that a dynamic exchange between organically bound deuterium and liquid water during water extraction can cause the stem water cryogenic extraction error, rather than deuterium fractionation occurs during root water uptake. Chen, Y., Helliker, B.R., Tang, X., Li, F., Zhou, Y. and Song, X. (2020) Stem water cryogenic extraction biases estimation in deuterium isotope composition of plant source water. Proceedings of the National Academy of Sciences, 202014422.

---

## Author Comment (AC1) · 30 Dec 2020

This paper addresses the use of isotopic data to partition evaporative water fluxes from terrestrial ecosystems, specifically focusing on transpiration vs. soil evaporation. It notes that isotopic methods have frequently been labelled "powerful," but have in fact proven difficult to use. The authors describe the barriers and complexities and propose several preferred pathways forward. It seems that these methods may be ready, at long last, to leap over the barriers that have constrained them. This paper prepares us for that leap by focusing attention on emerging best practices.

The topic is deeply relevant to BG because water fluxes control so many biogeochemical processes. This is so because, first, transpiration and soil evaporation are controlled by different environmental variables, but also because the water is drawn from different depths in the soil, which results in different vertical patterns of soil moisture. Third, the resulting moisture profiles affect many biogeochemical processes and fourth, the water fluxes redistribute solutes. It seems that these methods may be ready, at long last, to leap over the barriers that have constrained them. This paper prepares us for that leap by focusing attention on emerging best practices.

————————

A: Dear Prof. Marshall, In name of my co-authors, I would like to thank you for the time and effort you put into reviewing our manuscript! You will find below a list of answers to your general and technical comments.

————————

The paper begins with a review of earlier work and finishes with suggestions about how to move forward. Its novelty lies in the detection and presentation of trends in the broader literature and in the identification of key results in recent papers. These key results are mostly methodological. Because of this structure, the paper does not so much come to novel conclusions as emphasize promising methods. It might be helpful to begin with a summary of where the large variation in T/ET comes from. Before the series of T/ET values in Section 2, it would be useful to note that some of this variation is probably real and that some of the variation is predictable, e.g., when soils range from wet to dry or canopies range from isolated seedlings to closely spaced mature plants. The range of T/ET estimates in Section 2 would then make more sense. Also, I understand that Section 2 is intended as a timeline, but I wonder if it could be provided a bit more narrative flow. In particular, topic sentences at the beginnings of the paragraphs would help, if this is possible.

————————

A: Section 2 was, as a matter of fact, constructed as a pure timeline to underline new developments in isotopic sampling, analysis, and data interpretation techniques from 1990 up until today. We will give it a bit more "rhythm" in our revised manuscript, e.g., by using topic sentences, when possible, as suggested.
* * *
I was puzzled by the fact the isotopic estimates of T/ET were not directly compared to other methods. This is especially surprising because (Sutanto et al., 2014), which is not cited, made this comparison and concluded that the isotopic data yielded lower estimates. It would seem that this discrepancy should be presented and discussed. This could also precede Section 2.
* * *
A: Our main goal was to give a technical overview of the ensemble of isotopic partitioning methods and underline their challenges and progresses, and this is why we chose not to question/compare the isotope-derived T/ET results with those of non-isotope techniques. In our revised version, we will however cite the opinion paper of Sutanto et al. (2014) and report its main findings in the introduction section, before Section 2, as suggested, to put T/ET isotopic findings in perspective.
* * *
An important strength of Section 3 lies in the text descriptions of the meanings and assumptions of the mathematical models that have been used in this literature. This is not an easy sort of writing, but these authors do it well. My only general criticism was that I found it difficult to determine which of these insights were drawn from previous work and which are new here.
* * *
A: Thank you! We will do as asked and better distinguish the existing studies and their results from our suggested improvements/changes, especially in Sections 3.1.2, 3.2.2,

and 3.3.2.
* * *
In section 3.2, the authors discuss the isotopic signal of evaporation, focusing almost exclusively on soil evaporation while ignoring evaporation from plant surfaces (interception). Canopy evaporation can consume a substantial part of precipitation and it has isotopic consequences. There is some literature on the latter topic that includes descriptions of its isotopic consequences. The Allen et al. review (Allen et al., 2017) is one place to start. Much of the interception literature comes from forest canopies, but the same processes occur in crop canopies (e.g, Zheng et al., 2019). I presume that the models presented here include interception as part of transpiration, but perhaps they simply neglect wetted canopies. In either case, the treatment of interception should be explained clearly. It will be especially important in long-term estimates, in dense canopies, and where rainfall is frequent and light. I suppose it must also be much more important in sprinkler irrigation than in ditch or drip irrigation. If it is a research gap, I would highlight it in hopes that it will be addressed.
* * *
A: The issue of interception (i.e., direct evaporation of precipitation/irrigation water from leaves' surfaces) was not addressed here, this is true! In our revised version, we will clearly say that this additional and intermittent water vapor source is not identified in the mixing equation nor quantified in the literature reviewed in our manuscript, therefore not treated here.
* * *
It is also critical to recognize the contribution of Braden-Behrens et al., (2019), who have applied eddy covariance techniques to water stable isotope data, as suggested by the authors. This would seem to fit around line 337, at the climax of the methods section.
* * *
A: Many thanks! We have found another preprint of Jelka Braden-Behrens et al. (2020) in BGD, which will perfectly fit and strengthen Section 3.1.2.
* * *
I would also suggest more caution regarding the Keeling-Plot technique. Like the other methods described here, it is easily abused. As the authors note, the method depends on three important assumptions: first is that the method can only work if there are two and only two– uniform water sources in the mixture. This can be problem along vertical canopy profiles, where the isotopic composition of evaporating water is likely to vary with the rooting depth of the different species and perhaps, with the humidity and isotopic composition of the atmosphere surrounding the leaves. Perhaps this problem is less severe in a crop monoculture than in mixed vegetation, but the issue should be pointed out. The second problem with Keeling plots is that regression tends to flatten models fitted to noisy data, leading to incorrect estimates of the y-intercept. Because of this issue, some authors in the CO2 literature have recommended using the method only if the data meet fairly stringent requirements for R2. A wide range may help provide a high R2, but it does not guarantee it. Finally, there should be some discussion about which regression method should be used for the fitting of Keeling plots (Pataki et al., 2003; Wehr & Saleska, 2017).
* * *
A: We are ourselves very cautious in our application of the Keeling Plot in the field, but surely, this should better transpire from our text Section 3.1.2 and Section 4. We will incorporate your valuable comments and cited references, thank you.
* * *
Specific Comments: Line 16, 55: is it "powerful?" The manuscript argues otherwise later, both in a brief statement on lines 810-813 and in its overall tone. More than that,

the Sutanto et al. (2014) review raises serious questions about this. I would drop the first two paragraphs and replace them with the general description of T/ET requested above.
* * *
A: There is indeed a definite contradiction here with the rest of the text, thanks for pointing this out! We will remove "powerful" from the introduction (as well as from the abstract). Furthermore we will drop the first two §as requested in our revised manuscript.
* * *
L67-70: the iss issue is important, but complicated, in part because it depends on one's objectives. I know these authors want to talk about this, but I would wait to raise it until later, when it can really be dealt with.
* * *
A: We wanted to list the two factors, which act on the isotopic difference $\delta$T-$\delta$E, namely the differences in boundary conditions acting on T and E, and differences in transpiring vs. evaporating states (i.e., reaching or not of ISS for T and E). Since one factor cannot alone explain the isotopic difference, we would still prefer to keep them both listed. However we will make special reference regarding ISS to section 3 (and especially section 3.3.2) L67-70.
* * *
L81: futuring is in sec 4, not 3, right?
* * *
A: Already in section 3, there is mention to future progresses; however, we will make this much clearer, i.e., better highlight the improvements suggested in the literature and also distinguish them from our own suggested ones (, where we focus on monitoring

methods in particular). Section 4 is partly a summary of Section 3; it is the place where we give our opinion regarding ways forward for a continuous and non-destructive assessment of T/ET. We will make this also clearer.

———————

L85: progress L98: "noticeably low" and L104: "exceptionally high." Do the authors doubt these estimates? This should be clarified either as these comments are made or, perhaps better, in a final summary statement about true values of T/ET. As noted above, this paper would be strengthened by a general statement about what values T/ET should take and by a statement of how well the isotopic estimates match the alternatives.

———————

A: Yes, we agree (please see our answer to your general comment above.).

———————

L150: the Péclet effect seems important here and it should be explained carefully. It is more than compartmentalization. The effect is described in a bit more detail on lines 640-641, but it is not named there.

———————

A: To our knowledge and from our literature review, there is no study, in which Péclet effect values were determined for the specific purpose of ET partitioning. This stems certainly from the fact that calculations imply steady state in the first place ($\delta$T=$\delta$stem_water). This is why we did not explain the Peclet effect in Section 3.3.1. To make things clearer, we will remove mention to it L150.

———————

L265: explain ambient vs. backgd. Could this be called instead canopy vs. troposphere, for example?

———————————

A: We would prefer keeping the broader terms "ambient" and "background" for the following reasons: for e.g., crops or grasslands, air may be drawn from above the canopy, therefore does not apply strictly to "canopy air"; the background air may be different than troposphere air. However, we recognize that both terms are currently not precisely defined L265. We will do this in our revised manuscript. Thank you.

———————————

L280: deltaET, not ET

———————————

A: Thank you! This will be revised.

———————————

L321: a plus or minus symbol missing? Also there is no earlier equation estimating C from a Keeling plot. If there were, you should cite it by number.

———————————

A: We have checked the expression for the slope C and did not find an error. Nevertheless, we will rewrite the sentence L321 to: "We note that, by assuming $\hat{j}\_\chi\_atm \approx \chi\_atm$, the expressions for $\delta$atm provided by the flux gradient and Keeling plot techniques are mathematically identical if C=$\chi$_bg ($\delta$_bg-$\delta$_ET ) R_std=s·R_std, with s, the Keeling plot slope.

———————————

L351-2: Not a strong diagnostic as the linear form can survive a linear change in either variable.

———————————

A: Yes, this is true! Thank you for your remark, which we will add to the text.

[Figure]

————————

L395: different footprint areas

————————

A: Thank you. This will be revised.

————————

L460: hatm is determined by TDR?

————————

A: Not by TDR, but by, e.g., capacitive sensing. Thank you for pointing the mistake out! This will be revised accordingly.

————————

L582: A useful way to conclude this list of complications and worries would be to compare the estimated fluxes to empirical data from chambers or weighing lysimeters. This would allow the reader to decide how well these models work. I would do it with a figure.

————————

A: Thank you. We will mention that chambers and semi-controlled conditions experimental setups (such as weighing lysimeters) provide means to test the validity and existence of the abovementioned hypotheses and complications.

————————

L719-720: this point is so important, but it is not clearly worded. I would say something like: "...assume ISS and hence treat $\delta$xyl as equal to $\delta$T. Although this assumption is probably justified for a daily integration, there is growing evidence that plants reach ISS only briefly in the course of a day, especially when environmental conditions change rapidly. Thus the analysis is greatly simplified by daily integration, if that is sufficient for

the study objectives." But perhaps the authors disagree?
* * *
A: The authors absolutely agree! Thank you for this suggestion. We will use it in L719-720
* * *
L763: point, not punctual
* * *
A: Thank you! ________________

---

## Author Comment (AC2) · 30 Dec 2020

The submitted manuscript reviewed the methods of measuring or estimating $\delta$E, $\delta$Tand $\delta$ET for identifying the possible challenges of these isotopic methods, and how they should progress in the future, especially novel non-destructive methods. While this study is meaningful for the use of isotopic partitioning methods, the current literature overview and methodological review did not offer enough supports for the forementioned objectives. However, before the paper can be published, a major revision for this manuscript will be needed. The general and specific comments are showed as follows:

[Figure]

—————————

A: Dear Referee, On behalf of the co-authors, I thank you for these comments! Please find below a list of answers and foreseen changes to our manuscript.

—————————

General comments: 1. Literature overview: A literature overview was listed in the section 2 as a timeline for highlighting the important progresses made over the past 30 years, however, it might be helpful to classify and summarize these literature. If possible, suggest to add the topic sentences at the beginnings of each paragraphs for understanding easily.

—————————

A: Section 2 was, as a matter of fact, constructed as a pure timeline to underline new developments in isotopic sampling, analysis, and data interpretation techniques from 1990 up until today. We will give it a bit more "rhythm" in our revised manuscript, e.g., by using topic sentences, when possible, as suggested by you and Prof. John Marshall.

—————————

2. Methodological review: Suggest adding a theoretical schematic diagram including the flux of soil evaporation (E), transpiration (T), evapotranspiration (ET), and their stable isotopic compositions, and the key points in the estimation of $\delta$E, $\delta$T, and $\delta$ET in an ecosystem for understanding the calculation principle clearly.

—————————

A: Thank you for the suggestion. We will propose a general figure supporting text L55-79 in the introduction section.

—————————

3. Methodological review: One of assumptions of Keeling plot approach is temporal

variations in the water vapor mixing ratio and $\delta V$ are caused only by ET. However, rainfall, entrainment process of air, and so on can influence the variation in $\delta V$ at hourly or daily time scales, and introduces biases in the Keeling plot estimates. Suggest to add the analysis of the related uncertainties for Keeling plot approach.
* * *
A: We will add to Section 3.1.2 these factors of variation of $\delta v$, thank you.
* * *
4. Methodological review: Suggest to add the comprehensive comparisons of Keeling plot, flux gradient and EC isotopic flux method according to the calculation principle. Keeling plot and flux gradient methods can be agreement under certain conditions, and both calculation principles of flux gradient and EC isotopic flux methods is micrometeorological theory, however the discrepancy still exist among them (e. g. Hu et al., 2020 etc.). Hu, Y., Xiao, W., Wei, Z., Welp, L., Wen, X., Lee, X. (2020) Determining the Isotopic Composition of Surface Water Vapor Flux From High-Frequency Observations Using Flux-Gradient and Keeling Plot Methods. Earth and Space Science, DOI: 10.1002/es-soar.10501239.1.
* * *
A: Our objective was to provide the (non-specialist) reader with an overview of the methods for determination of $\delta ET$, $\delta E$, $\delta T$ and shortly highlight differences among them. An in-depth comparison of the micrometeorological methods for determination of $\delta ET$ seems out of focus here. Nevertheless, we will refer the reader, e.g., cite the above study, for more details on this.
* * *
5. Methodological review: How about the uncertainty analysis of EC isotopic flux methods due to the loss of information during the covariance calculation between the isotopic compositions and vertical wind fluctuations?

—————————

A: Thank you. We will add this very important technical aspect section 3.1.2

—————————

6. Methodological review: Suggest to clarifying the calculations and assumptions of converting $\delta$ value of water vapor into $\delta$ value of soil or xylem (liquid) water, and the analysis of possible uncertainty in detailed, for promoting the use of the novel non-destructive methods in the future.

—————————

A: As stated L514-518, all authors simply assume steady state between both liquid and vapour phases. Consequently, only temperature is needed to convert $\delta\_soil^{vap}$ into $\delta\_soil^{liq}$ values, e.g., using the equations of Majoube (1971) and Horita and Wesolowski (1994). We further note in the text that only Rothfuss et al. (2013) provided evidence of near-isotopic equilibrium conditions between liquid and vapour in the soil pore space.

—————————

3. Possible ways forward: It might be useful to focus on the improvement of difficult problems of recommended methods, for example, how the plant-size measurements based on the chamber methods are scaled up for canopy-level estimations.

—————————

A: We argue L788-792 that chamber measurements should be replicated in space to "characterize the in situ natural lateral heterogeneity of $\delta$T, due to differences in root water uptake, plant physiological state, as well as lateral heterogeneity in soil water isotopic composition profiles". This is a prerequisite for any upscaling attempt. We will make this clearer in the text, thank you!

—————————

Specific comments: Line 405: if the describe of "the second relies on destructive sampling of the soil and offline analysis of the extracted water" is suitable, and the non-destructive collection and online monitoring of the water vapour isotopic composition of soil atmosphere also can be used.
* * *
A: This is true, thank you! This will be revised accordingly.
* * *
Line 640-670: Péclet effect is very important for these theoretical equations, however, its definition and quantification are not explicitly named here.
* * *
A: To our knowledge and from our literature review, there is no study, in which Péclet effect values were determined for the specific purpose of ET partitioning. This stems certainly from the fact that calculations imply steady state in the first place ($\delta$T=$\delta$stem_water). This is why we did not explain the Peclet effect in Section 3.3.1. To make things clearer, we will remove mention to it L150 and not mention it in the manuscript.
* * *
Line 735: For the cryogenic extraction methods of xylem water, new biases might be considered, because Chen et al. (2020) found that a dynamic exchange between organically bound deuterium and liquid water during water extraction can cause the stem water cryogenic extraction error, rather than deuterium fractionation occurs during root water uptake.

Chen, Y., Helliker, B.R., Tang, X., Li, F., Zhou, Y.and Song, X. (2020) Stem water cryogenic extraction biases estimation in deuterium isotope composition of plant source water. Proceedings of the National Academy of Sciences, 202014422
* * *
A: We thank you for this nice new paper on issues related with extraction of water from plant tissues and will not omit citing it in our revised manuscript!

References

Horita, J., and Wesolowski, D. J.: Liquid-vapor fractionation of oxygen and hydrogen isotopes of water from the freezing to the critical-temperature, Geochim. Cosmochim. Acta, 58, 3425-3437, https://doi.org/10.1016/0016-7037(94)90096-5, 1994.

Majoube, M.: Oxygen-18 and deuterium fractionation between water and steam, J. Chim. Phys. Phys.-Chim. Biol., 68, 1423-&, 1971.

---

## Editor Decision (ED1)

**Additional changes requested for MS bg-2020-414:**

**Referee #1**

Deal with canopy interception in some form to make this analysis of T/ET complete. Suggestion: add a point v) to the summary, which states that the methods will not apply broadly until they are able to deal with canopy interception. This point provides critical context for the good work contained in this review.

**Referee #2**

(1) Péclet effect may affect the simulation of δT using the case scenarios (i) and (ii) with the assumption that leaf water is a well-mixed reservoir, and further affect T/ET values. This effect was described on lines 675-678 but it is not named there, although you removed mention to Péclet effect in Literature overview.
(2) Please check and correct the grammar and spelling, eg. "was the not the major focus" in Line 94-95.

**Editors comment:**

Line 888: 'punctual' is probably not the right word here. Maybe discontinuous or sporadic?

---

## Author Response (AR2)

**Additional changes requested for MS bg-2020-414:**

Dear Dr. Trebs,

On behalf of my co-authors, I again thank you for the swift handling of our manuscript. We hope to have addressed the technical comments of both reviewers below in a satisfying way

Sincerely,

Youri Rothfuss

**Referee #1**

Deal with canopy interception in some form to make this analysis of T/ET complete. Suggestion: add a point v) to the summary, which states that the methods will not apply broadly until they are able to deal with canopy interception. This point provides critical context for the good work contained in this review.

Dear Prof. Marshall,

We added the following text at the very end of the Summary section (L851-856). We hope it will sufficiently underline the limits of the two-end member partitioning equation concept, which neglects interception:

"*Importantly, and to conclude this summary, the general isotopic partitioning approach (i.e., Eq. (1)) as well as the ensemble of methods and their possible improvements will not be applied broadly until they are able to deal with canopy interception. Further research is therefore needed to (i) determine the water volumes collected by the vegetation following rain events, fog deposition, or dew condensation and to (i) investigate the isotopic effects during re-evaporation of the intercepted water. This should be useful for constraining a generalized partitioning equation including a third member, namely the stable isotopic composition of interception.*"

**Referee #2**

(1) Péclet effect may affect the simulation of δT using the case scenarios (i) and (ii) with the assumption that leaf water is a well-mixed reservoir, and further affect T/ET values. This effect was described on lines 675-678 but it is not named there, although you removed mention to Péclet effect in Literature overview.

Dear anonymous Referee,

The studies cited lines 675-678 reported observations on leaf water isotopic composition heterogeneity but did not make use of the Péclet number (as none of the reviewed literature did), rather tested the series of Equations (27-30). As you point out, we do acknowledge the importance of a non-well-mixed leaf water reservoir in the calculation of the value of transpiration isotopic composition and ultimately its impact on T/ET estimate (L673-710).

(2) Please check and correct the grammar and spelling, eg. "was the not the major focus" in Line 94-95.

Thank you: we removed this part of the sentence for ease of reading.

**Editors comment:**

Line 888: 'punctual' is probably not the right word here. Maybe discontinuous or sporadic?

Dear Dr. Trebs,

We replaced "punctual" by "*discontinuous*" (L805). Thank you for this suggestion!

**Comment from Editor Dr. Ivonne Trebs**

Dear Prof. Dr. Rothfuss,

Thank you for submitting your manuscript "Reviews and syntheses: Gaining insights into evapotranspiration partitioning with novel isotopic monitoring methods" to Biogeosciences. After reviewing the comments made by the referees and your response letters, I find that your manuscript requires major revisions to address the main concerns of the referees. Your responses to the referee comments and suggested changes to the paper are very detailed and complete. Please incorporate these into the revised version of the manuscript.

Personally, I do not agree on the exclusion of interception evaporation, even though there are methodological constraints as you indicated in your response letters. I would appreciate if this topic will be discussed at least shortly in the paper. As the paper addresses a broader readership it should be clearly stated that the main equations and methods are only valid for soil evaporation/transpiration partitioning.

Best regards,

Ivonne Trebs

> Dear Dr. Trebs,
>
> On behalf of my co-authors, I would like to thank you for coordinating the review of our manuscript! We hope to have addressed all comments from Prof. Marshall and one anonymous referee satisfyingly.
>
> The issue of interception (i.e., direct evaporation of precipitation/irrigation water from leaves' surfaces) was indeed not mentioned in our initial submission. We now fill this gap in the introduction (L61-65):
>
> "*The spatiotemporal variabilities of these factors and the complexity of their interactions may result in significant heterogeneous distributions of both $\delta_E$ and $\delta_T$ in the field (Fig. 1). Importantly and as reflected by the reviewed isotopic literature (see Section 2), E in this context does not include canopy interception and dew evaporation, which are known to be associated with isotopic effects (Allen et al., 2017; Zheng et al., 2019). Theses fluxes can be of significant magnitude, depending on the scale of interest (Good et al., 2015; Allen et al., 2017).*"
>
> Please find below a list of our answers to the referees general and technical comments.
>
> Kind Regards from Jülich,
>
> Youri Rothfuss

**Review of Referee #1 Prof. John Marshall**

This paper addresses the use of isotopic data to partition evaporative water fluxes from terrestrial ecosystems, specifically focusing on transpiration vs. soil evaporation. It notes that isotopic methods have frequently been labelled "powerful," but have in fact proven difficult to use. The authors describe the barriers and complexities and propose several preferred pathways forward. It seems that these methods may be ready, at long last, to leap over the barriers that have constrained them. This paper prepares us for that leap by focusing attention on emerging best practices.

The topic is deeply relevant to BG because water fluxes control so many biogeochemical processes. This is so because, first, transpiration and soil evaporation are controlled by different environmental variables, but also because the water is drawn from different depths in the soil, which results in different vertical patterns of soil moisture. Third, the resulting moisture profiles affect many biogeochemical processes and fourth, the water fluxes redistribute solutes. It seems that these methods may be ready, at long last, to leap over the barriers that have constrained them. This paper prepares us for that leap by focusing attention on emerging best practices.

> Dear Prof. Marshall,
>
> In name of the co-authors, I would like to thank you for the time and effort you put into reviewing our manuscript! You will find below a list of answers to your general and technical comments.

The paper begins with a review of earlier work and finishes with suggestions about how to move forward. Its novelty lies in the detection and presentation of trends in the broader literature and in the identification of key results in recent papers. These key results are mostly methodological. Because of this structure, the paper does not so much come to novel conclusions as emphasize promising methods.

> This was, indeed, our intention: focus on promising methods, while summarizing the existing ones thoroughly to motivate researchers willing to use water stable isotopes for the specific purpose of ET partitioning.

It might be helpful to begin with a summary of where the large variation in T/ET comes from. Before the series of T/ET values in Section 2, it would be useful to note that some of this variation is probably real and that some of the variation is predictable, e.g., when soils range from wet to dry or canopies range from isolated seedlings to closely spaced mature plants. The range of T/ET estimates in Section 2 would then make more sense.

> We have added a new Figure 1 to the manuscript (as also suggested by Referee #2 request) illustrating the possible high variabilities of the environmental factors driving $\delta_E$ and $\delta_T$. Figure 1 should also highlight the difficulty of applying the isotopic methodology to mixed-vegetation covers.

[Figure]

*"Figure 1: Conceptual drawing reporting the sources of differences in (synthetic) values between the (exemplary oxygen) isotopic compositions of evaporation ($\delta_E$ [‰]) and transpiration ($\delta_T$ [‰]) in an agroforestry context, namely (i) the type of vegetation and root development (tree vs. maize crop vs. grass layer), (ii) the prevalence of isotopic steady state (ISS) or non-steady state (NSS) conditions for leaf water, and (iii) the environmental conditions acting on fluxes, i.e., soil water and atmospheric water vapour isotopic composition profiles, and leaf water isotopic composition (values displayed in brown, blue, and green outlined boxes). $\delta_E$ and NSS $\delta_T$ values were calculated with the Craig and Gordon (1965) model assuming laminar flow conditions (designated by the three superimposed arrows) under the pictured tree and within its canopy and fully turbulent conditions (designated by a circular arrow) elsewhere (e.g., at the top of the tree canopy, and above the maize crop for $\delta_T$ and in its interrow space for $\delta_E$). pictured tree and within its canopy and fully turbulent conditions elsewhere (e.g., above the maize crop for $\delta_T$ and in its interrow space for $\delta_E$)."*

Also, I understand that Section 2 is intended as a timeline, but I wonder if it could be provided a bit more narrative flow. In particular, topic sentences at the beginnings of the paragraphs would help, if this is possible.

> Section 2 was, as a matter of fact, constructed as a pure timeline to underline new developments in isotopic sampling, analysis, and data interpretation techniques from 1990 up until today. We understand it might be difficult to read this part of the text due to its 'linear' nature. We have now pooled publications by topics in separate paragraphs, e.g., studies investigating *vegetation type or strata-specific transpiration to evapotranspiration ratio* or *the impact of irrigation on the partitioning of ET*, studies *working with a common water isotope mass-balance equation*, studies that have *simulated the T/ET using Isotope-enabled, physically-based, and numerically-solved soil-vegetation-atmosphere transfer*.
>
> We hope that it is now friendlier to read!

I was puzzled by the fact the isotopic estimates of T/ET were not directly compared to other methods. This is especially surprising because (Sutanto et al., 2014), which is not cited, made

this comparison and concluded that the isotopic data yielded lower estimates. It would seem that this discrepancy should be presented and discussed. This could also precede Section 2.

> Our main goal was to give a technical overview of the ensemble of isotopic partitioning methods and underline their challenges and progresses, and this is why we chose not to question/compare the isotope-derived T/ET results with those of non-isotope techniques. We now cite the opinion paper of Sutanto et al. (2014) and report its main findings in the introduction section, before Section 2, as suggested, to put T/ET isotopic findings in perspective (L80-82):
>
> "*Note also that this study will not focus on differences in T/ET as estimated by the abovementioned traditional methods on the one hand and by the isotopic methods on the other; this has been extensively reported by, e.g., Sutanto et al. (2014).*"
>
> We now also mention in Section 2, L200-202, the study of Sutanto et al. (2014) for introducing the paragraph relating to studies, which have focused on the comparison between isotopic and non-isotopic estimation of T/ET:
>
> "*Sutanto et al. (2014) reported from the literature generally higher isotope-derived T/ET (> 70 %) values than those of the traditional approaches for comparable land cover types. However, at experimental sites combining both type of measurements, Sutanto et al. (2014) underlined a fair agreement between both approaches.*"

An important strength of Section 3 lies in the text descriptions of the meanings and assumptions of the mathematical models that have been used in this literature. This is not an easy sort of writing, but these authors do it well. My only general criticism was that I found it difficult to determine which of these insights were drawn from previous work and which are new here.

> Thank you! We tried to better distinguish the existing studies and their results from our suggested improvements/changes, especially in Sections 3.1.2, 3.2.2, and 3.3.2. Especially in Section 3.2.2, we now cite our previous work, where, in the original submission, it "looked" like we were presenting new insights. In addition, we now specify the difference between these section on the one hand (L76-78):
>
> "*The central aim of this study is to identify from the literature the challenges the ensemble of isotopic methods currently face and how they should progress in the future (section 3)*"
>
> and Section 5 on the other hand (L84-85):
>
> "*Finally, section 5 presents a summary of our own suggestion for improvement as well as of the possible ways forward for the isotopic partitioning community.*"

In section 3.2, the authors discuss the isotopic signal of evaporation, focusing almost exclusively on soil evaporation while ignoring evaporation from plant surfaces (interception). Canopy evaporation can consume a substantial part of precipitation and it has isotopic consequences. There is some literature on the latter topic that includes descriptions of its isotopic consequences. The Allen et al. review (Allen et al., 2017) is one place to start. Much of the interception literature comes from forest canopies, but the same processes occur in crop canopies (e.g, Zheng et al., 2019). I presume that the models presented here include interception as part of transpiration, but perhaps they simply neglect wetted canopies. In either case, the treatment of interception should be explained clearly. It will be especially important in long-term estimates, in dense canopies, and where rainfall is frequent and light. I suppose it must also be much more important in sprinkler irrigation than in ditch or drip irrigation. If it is a research gap, I would highlight it in hopes that it will be addressed.

> The issue of interception (i.e., direct evaporation of precipitation/irrigation water from leaves' surfaces) was not addressed here, this is true! It is now mentioned in the introduction as (L62-65):

> *"Importantly and as reflected by the reviewed isotopic literature (see Section 2), E in this context does not include canopy interception and dew evaporation, which are known to be associated with isotopic effects (Allen et al., 2017; Zheng et al., 2019). Theses fluxes can be of significant magnitude, depending on the scale of interest (Good et al., 2015; Allen et al., 2017)."*

It is also critical to recognize the contribution of Braden-Behrens et al., (2019), who have applied eddy covariance techniques to water stable isotope data, as suggested by the authors. This would seem to fit around line 337, at the climax of the methods section.

> Many thanks! We have found another preprint of Jelka Braden-Behrens et al. (2020) in BGD, which perfectly fit and strengthen Section 3.1.2. It now reads (L427-430):

> *"Recently, Braden-Behrens et al. (2019; 2020) showed that EC measurements could be performed using a high flow ($\varphi \approx 4.2$ l min-1) laser spectrometer clocked at 2 Hz only (2 Hz-HF-WVIA, Los Gatos Research Inc., San Jose, CA, USA). They underlined the importance of heating the sampling tubing at the point of intake in order to avoid problems of condensation and high-frequency dampening as showed from spectral and cospectral analyses"*

I would also suggest more caution regarding the Keeling-Plot technique. Like the other methods described here, it is easily abused. As the authors note, the method depends on three important assumptions: first is that the method can only work if there are two and only two– uniform water sources in the mixture. This can be problem along vertical canopy profiles, where the isotopic composition of evaporating water is likely to vary with the rooting depth of the different species and perhaps, with the humidity and isotopic composition of the atmosphere surrounding the leaves. Perhaps this problem is less severe in a crop monoculture than in mixed vegetation, but the issue should be pointed out. The second problem with Keeling plots is that regression tends to flatten models fitted to noisy data, leading to incorrect estimates of the y-intercept. Because of this issue, some authors in the CO2 literature have recommended using the method only if the data meet fairly stringent requirements for R2. A wide range may help provide a high R2, but it does not guarantee it. Finally, there should be some discussion about which regression method should be used for the fitting of Keeling plots (Pataki et al., 2003; Wehr & Saleska, 2017).

> Thank your for your comment. We have substantially revised the part on the Keeling plot technique in our revised manuscript and we hope that it strengthened Section 3.1.2 significantly (L349-360):

> *"In a review of isotope techniques for determination of the concomitant flux and isotopic composition of evapotranspiration, Griffis (2013) summarized the inherent limitations of the Keeling plot technique from the literature. The general assumption that atmospheric water vapour and its isotopic composition result from the turbulent mixing of only two sources was reported to be often violated. Reasons for this may be strong vertical gradients of water vapour mixing ratio and isotopic composition or strong differences between $\delta E$ and $\delta T$ leading to the emergence of diffusion and air entrainment processes (Lee et al., 2006; Lee et al., 2012). The 'spatial' Keeling plot approach suffers particularly from the fact that the different heights at which the $\delta_{atm}$ is measured are representative for differently footprints areas of the studied ecosystem. While this may not be a problem for a homogeneous cropland, the reliability of the Keeling plot should be generally questioned for a mixed vegetation (such as represented in Fig. 1) with strong lateral variabilities in $\delta_{atm}$ and $\chi_{atm}$, but also in soil water isotopic composition. In addition, the application of the spatial Keeling plot should not be conditioned based on a wide span of $\chi_{atm}$ values only but naturally on the quality of its linear fit. Griffis (2013) argued as well that the flux gradient approach suffers from a narrow range of application, e.g., may not be suitable in certain cases, such as below forest canopies or above tall vegetation. "*

Specific Comments:

- Line 16, 55: is it "powerful?" The manuscript argues otherwise later, both in a brief statement on lines 810-813 and in its overall tone. More than that, the Sutanto et al. (2014) review raises serious questions about this. I would drop the first two paragraphs and replace them with the general description of T/ET requested above.
  - There is indeed a definite contradiction here with the rest of the text, thanks for pointing this out! We have removed "powerful" from the introduction (as well as from the abstract). Furthermore we have dropped the first two § as requested in our revised manuscript.

- L67-70: the iss issue is important, but complicated, in part because it depends on one's objectives. I know these authors want to talk about this, but I would wait to raise it until later, when it can really be dealt with.
  - We wanted to list the two factors, which act on the isotopic difference δT-δE, namely the differences in boundary conditions acting on T and E, and differences in transpiring vs. evaporating states (i.e., reaching or not of ISS for T and E). Since one factor cannot alone explain the isotopic difference, we would still prefer to keep them both listed. However we made special reference regarding ISS to section 3 (and especially section 3.3.2) L57-58:
    "(ii) the prevalence (or non-prevalence) of isotopic steady state (ISS) for transpiration, i.e. whether $\delta_T$ is independent of time (Farquhar and Cernusak, 2005; Dubbert et al., 2014a) (**see Section 3 for a detailed description of ISS**)."

- L81: futuring is in sec 4, not 3, right?
  - Please see our answer to your general comment above.

- L85: progress L98: "noticeably low" and L104: "exceptionally high." Do the authors doubt these estimates? This should be clarified either as these comments are made or, perhaps better, in a final summary statement about true values of T/ET. As noted above, this paper would be strengthened by a general statement about what values T/ET should take and by a statement of how well the isotopic estimates match the alternatives.
  - Yes, we agree (please see our answer to your general comment above.).

- L150: the Péclet effect seems important here and it should be explained carefully. It is more than compartmentalization. The effect is described in a bit more detail on lines 640-641, but it is not named there.
  - Peclet effect (PE) is indeed important for quantifying the heterogeneous distribution of $\delta_L$. However, and after careful review, we did not find a study, in which PE was used to simulate $\delta_T$ and ultimately T/ET values. Dubbert et al., (2014) mentions PE as hint upon discrepancies between $\delta_L$ measurements and simulations while Wang et al. (2015) test different $\delta_L$ models against each other over the course of one day. Piayda et al. (2017) presents the PE however do not report the results, arguing that "small differences in isotopic composition were found [between the bulk leaf water and water at the evaporative sites], which were not significant for the results shown in [their] work". This is why we did not review the use of PE presently.
  In order to avoid possible confusion, we removed mention to PE L150.

    Dubbert, M., Piayda, A., Cuntz, M., Correia, A. C., Costa E Silva, F., Pereira, J. S., and Werner, C.: Stable oxygen isotope and flux partitioning demonstrates understory of an oak savanna contributes up to half of ecosystem carbon and water exchange, Front Plant Sci, 5, 530, doi:10.3389/fpls.2014.00530, 2014.
    Wang, P., Yamanaka, T., Li, X. Y., and Wei, Z. W.: Partitioning evapotranspiration in a temperate grassland ecosystem: Numerical modeling with isotopic tracers, Agr. Forest Meteorol., 208, 16-31, doi:10.1016/j.agrformet.2015.04.006, 2015.
    Piayda, A., Dubbert, M., Siegwolf, R., Cuntz, M., and Werner, C.: Quantification of dynamic soil-vegetation feedbacks following an isotopically labelled precipitation pulse, Biogeosciences, 14, 2293-2306, doi:10.5194/bg-14-2293-2017, 2017.

- L265: explain ambient vs. backgd. Could this be called instead canopy vs. troposphere, for example?
  - We would prefer keeping the broader terms "ambient" and "background" for the following reasons: for e.g., crops or grasslands, air may be drawn from above the canopy, therefore does not apply strictly to "canopy air"; the background air may be different than troposphere air. However, we recognize that both terms were not precisely defined L265 in the original submission. We now do this L264-267:
  "*The so-called 'Keeling plot' technique simply considers that the water vapour measured in some ecosystem atmosphere (of concentration Catm, dimension of $M\ L^{-3}$), e.g., within or above the canopy, originates from two sources, namely (i) the background water vapour (of concentration $C_{bg}\ [M\ L^{-3}]$), transported advectively and defined as not being influenced by ET flux, and (ii) evapotranspiration ET (of concentration $C_{ET}\ [M\ L^{-3}]$):*"

- L280: deltaET, not ET
  - Thank you! This is now revised.

- L321: a plus or minus symbol missing? Also there is no earlier equation estimating C from a Keeling plot. If there were, you should cite it by number.
  - We have checked the expression for the slope C and did not find an error... Nevertheless, we rewrote the sentence (now L329-331) as:
  "*We note that, by assuming $^{j}\chi_{atm} \approx \chi_{atm}$, the flux gradient and Keeling plot techniques are mathematically identical if $C = \chi_{bg}(\delta_{bg} - \delta_{ET})R_{std} = s \cdot R_{std}$, with s, the Keeling plot slope.*

- L351-2: Not a strong diagnostic as the linear form can survive a linear change in either variable.
  - Yes, this is true! Thank you for your remark, which was added to the text (L380-381).

- L395: different footprint areas
  - Thank you. This was revised (L354-355):
  "*The 'spatial' Keeling plot approach suffers particularly from the fact that the different heights at which the $\delta_{atm}$ is measured are representative for differently footprints areas of the studied ecosystem.*"

- L460: hatm is determined by TDR?
  - Not by TDR, but by, e.g., capacitive sensing. Thank you for pointing the mistake out! This is revised accordingly (L493-494):
  "*The first two variables are typically monitored with, e.g., capacitive sensing*"

- L582: A useful way to conclude this list of complications and worries would be to compare the estimated fluxes to empirical data from chambers or weighing lysimeters. This would allow the reader to decide how well these models work. I would do it with a figure.
  - Thank you. We now mention that chambers and semi-controlled conditions experimental setups (such as weighing lysimeters) provide means to test the validity and existence of the abovementioned hypotheses and complications (L616-618):
  "*Gas-exchange chambers and other experimental setups with semi-controlled conditions (such as weighing lysimeters) provide means to test the validity and*

> *existence of the abovementioned hypotheses and complications (e.g., Dubbert et al., 2013; Groh et al., 2018)"*

- L719-720: this point is so important, but it is not clearly worded. I would say something like: "...assume ISS and hence treat δxyl as equal to δT. Although this assumption is probably justified for a daily integration, there is growing evidence that plants reach ISS only briefly in the course of a day, especially when environmental conditions change rapidly. Thus the analysis is greatly simplified by daily integration, if that is sufficient for the study objectives." But perhaps the authors disagree?
  - The authors absolutely agree! Thank you for this suggestion, which we incorporated in the manuscript (L764-767):
    *"A number of studies (e.g., Zhou et al., 2018; Wei et al., 2015; Aouade et al., 2016; Volkmann et al., 2016) assume ISS and hence treat δxyl as equal to δT, Although this assumption is probably justified for a daily integration, there is growing evidence that plants reach ISS only briefly in the course of a day, especially when environmental conditions change rapidly (Simonin et al., 2013; Dubbert et al., 2014b; 2017)."*
- L763: point, not punctual
  - Thank you!

**Review of Referee #2**

The submitted manuscript reviewed the methods of measuring or estimating δE, δTand δET for identifying the possible challenges of these isotopic methods, and how they should progress in the future, especially novel non-destructive methods. While this study is meaningful for the use of isotopic partitioning methods, the current literature overview and methodological review did not offer enough supports for the forementioned objectives. However, before the paper can be published, a major revision for this manuscript will be needed. The general and specific comments are showed as follows:

> Dear Referee,
>
> On behalf of the co-authors, I thank you for these comments! Please find below a list of answers and foreseen changes to our manuscript.

General comments:

1. Literature overview: A literature overview was listed in the section 2 as a timeline for highlighting the important progresses made over the past 30 years, however, it might be helpful to classify and summarize these literature. If possible, suggest to add the topic sentences at the beginnings of each paragraphs for understanding easily.

> Section 2 was, as a matter of fact, constructed as a pure timeline to underline new developments in isotopic sampling, analysis, and data interpretation techniques from 1990 up until today. We understand it might be difficult to read this part of the text due to its 'linear' nature. We have now pooled publications by topics in separate paragraphs, e.g., studies investigating *vegetation type or strata-specific transpiration to evapotranspiration ratio* or *the impact of irrigation on the partitioning of ET*, studies *working with a common water isotope mass-balance equation*, studies that have *simulated the T/ET using Isotope-enabled, physically-based, and numerically-solved soil-vegetation-atmosphere transfer*.
>
> We hope that it is now friendlier to read!

2. Methodological review: Suggest adding a theoretical schematic diagram including the flux of soil evaporation (E), transpiration (T), evapotranspiration (ET), and their stable isotopic compositions, and the key points in the estimation of δE, δT, and δET in an ecosystem for understanding the calculation principle clearly.

> Thank you for the suggestion. We have added a new Figure 1 to the manuscript (as also suggested by Referee #2 request) illustrating the possible high variabilities of the environmental factors driving $\delta_E$ and $\delta_T$. Figure 1 should also highlight the difficulty of applying the isotopic methodology to mixed-vegetation covers.

[Figure]

*"Figure 1: Conceptual drawing reporting the sources of differences in (synthetic) values between the (exemplary oxygen) isotopic compositions of evaporation ($\delta_E$ [‰]) and transpiration ($\delta_T$ [‰]) in an agroforestry context, namely (i) the type of vegetation and root development (tree vs. maize crop vs. grass layer), (ii) the prevalence of isotopic steady state (ISS) or non-steady state (NSS) conditions for leaf water, and (iii) the environmental conditions acting on fluxes, i.e., soil water and atmospheric water vapour isotopic composition profiles, and leaf water isotopic composition (values displayed in brown, blue, and green outlined boxes). $\delta_E$ and NSS $\delta_T$ values were calculated with the Craig and Gordon (1965) model assuming laminar flow conditions (designated by the three superimposed arrows) under the pictured tree and within its canopy and fully turbulent conditions (designated by a circular arrow) elsewhere (e.g., at the top of the tree canopy, and above the maize crop for $\delta_T$ and in its interrow space for $\delta_E$). pictured tree and within its canopy and fully turbulent conditions elsewhere (e.g., above the maize crop for $\delta_T$ and in its interrow space for $\delta_E$)."*

3. Methodological review: One of assumptions of Keeling plot approach is temporal variations in the water vapor mixing ratio and δV are caused only by ET. However, rainfall, entrainment process of air, and so on can influence the variation in δV at hourly or daily time scales, and introduces biases in the Keeling plot estimates. Suggest to add the analysis of the related uncertainties for Keeling plot approach.

*Thank your for your comment. We have substantially revised the part on the Keeling plot technique in our revised manuscript and we hope that it strengthened Section 3.1.2 significantly (L349-360):*

*"In a review of isotope techniques for determination of the concomitant flux and isotopic composition of evapotranspiration, Griffis (2013) summarized the inherent limitations of the Keeling plot technique from the literature. The general assumption that atmospheric water vapour and its isotopic composition result from the turbulent mixing of only two sources was reported to be often violated. Reasons for this may be strong vertical gradients of water vapour mixing ratio and isotopic composition or strong differences between $\delta_E$ and $\delta_T$ leading to the emergence of diffusion and air entrainment processes (Lee et al., 2006; Lee et al., 2012). The 'spatial' Keeling plot approach suffers particularly from the fact that the different heights at which the $\delta_{atm}$ is measured are*

*representative for differently footprints areas of the studied ecosystem. While this may not be a problem for a homogeneous cropland, the reliability of the Keeling plot should be generally questioned for a mixed vegetation (such as represented in Fig. 1) with strong lateral variabilities in $\delta_{atm}$ and $\chi_{atm}$, but also in soil water isotopic composition. In addition, the application of the spatial Keeling plot should not be conditioned based on a wide span of $\chi atm$ values only but naturally on the quality of its linear fit. Griffis (2013) argued as well that the flux gradient approach suffers from a narrow range of application, e.g., may not be suitable in certain cases, such as below forest canopies or above tall vegetation.* "

4. Methodological review: Suggest to add the comprehensive comparisons of Keeling plot, flux gradient and EC isotopic flux method according to the calculation principle. Keeling plot and flux gradient methods can be agreement under certain conditions, and both calculation principles of flux gradient and EC isotopic flux methods is micrometeorological theory, however the discrepancy still exist among them (e. g. Hu et al., 2020 etc.).

Hu, Y., Xiao, W., Wei, Z., Welp, L., Wen, X., Lee, X. (2020) Determining the Isotopic Composition of Surface Water Vapor Flux From High-Frequency Observations Using Flux-Gradient and Keeling Plot Methods. Earth and Space Science, DOI: 10.1002/es-soar.10501239.1.

> Our objective was to provide the (non-specialist) reader with an overview of the methods for determination of $\delta_{ET}$, $\delta_E$, $\delta_T$ and shortly highlight differences among them. An in-depth comparison of the micrometeorological methods for determination of $\delta_{ET}$ seemed to us out of focus here. Nevertheless, we now refer the reader to the study of Hu et al. (2020) and shortly summarize their findings (L370-376):
>
> "*Hu et al. (2020) compared at one irrigated maize crop δET values determined with either the Keeling plot or the flux gradient approaches. They tested different regression models with the Keeling plot method, i.e., ordinary least squares regression, geometric mean regression, and York's solution (see for details, Pataki et al., 2003; Wehr and Saleska, 2017). These models differ in the way errors made on 1/χatm and δatm (see Eq. (5)) relate to each other and whether they may be considered as constant over their measurement ranges. As such, they yield differences in δET estimates. Hu et al. (2020) could illustrate the necessity of choosing an appropriate regression model that reflects the dependency of spectrometer-specific errors on water vapour mixing ratio.*"

5. Methodological review: How about the uncertainty analysis of EC isotopic flux methods due to the loss of information during the covariance calculation between the isotopic compositions and vertical wind fluctuations?

> Thank you. We added this very important technical aspect L415-419:
>
> "*Compared to the Keeling plot and flux-gradient approaches, the eddy covariance technique derives from micrometeorological theory (first principles). Where applicable, this makes it a solid alternative less subjected to assumptions. However, and as a result of its high data acquisition rate and associated noise, the EC technique provides δ$_{ET}$ estimates with higher uncertainty, largely determined by random measurement errors (Hollinger and Richardson, 2005; Loescher et al., 2006; Rannik et al., 2016). Good et al. (2012) determined this uncertainty to be proportional to the inverse of the correlation coefficient between ω and χatm, i.e., the covariance of ω and χatm divided by the product of their measurement errors.*"

6. Methodological review: Suggest to clarifying the calculations and assumptions of converting δ value of water vapor into δ value of soil or xylem (liquid) water, and the analysis of possible uncertainty in detailed, for promoting the use of the novel non-destructive methods in the future.

> As stated now L548-551, all authors simply assume steady state between both liquid and vapour phases. Consequently, only temperature is needed to convert $\delta_{\text{soil}}^{\text{vap}}$ into $\delta_{\text{soil}}^{\text{liq}}$

values, e.g., using the equations of Majoube (1971) and Horita and Wesolowski (1994). We further note in the text that only Rothfuss et al. (2013) provided evidence of near-isotopic equilibrium conditions between liquid and vapour in the soil pore space.

Horita, J., and Wesolowski, D. J.: Liquid-vapor fractionation of oxygen and hydrogen isotopes of water from the freezing to the critical-temperature, Geochim. Cosmochim. Acta, 58, 3425-3437, https://doi.org/10.1016/0016-7037(94)90096-5, 1994.

Majoube, M.: Oxygen-18 and deuterium fractionation between water and steam, J. Chim. Phys. Phys.-Chim. Biol., 68, 1423-&, 1971.

Rothfuss, Y., Vereecken, H., and Brüggemann, N.: Monitoring water stable isotopic composition in soils using gas-permeable tubing and infrared laser absorption spectroscopy, Water Resour. Res., 49, 1-9, doi:10.1002/wrcr.20311, 2013.

3. Possible ways forward: It might be useful to focus on the improvement of difficult problems of recommended methods, for example, how the plant-size measurements based on the chamber methods are scaled up for canopy-level estimations.

We argue now L830-833 that chamber-based or leaf-based measurements should be replicated in space to support any upscaling attempt of $\delta_T$:

"*(iii) several transparent flushed plant or leaf-size chambers should be operated at the study site to characterize the in situ natural lateral heterogeneity of $\delta_T$, due to differences in root water uptake, plant physiological state, as well as lateral heterogeneity in soil water isotopic composition profiles. This would be a prerequisite for any upscaling attempt of $\delta_T$ values.*"

Specific comments:

1. Line 405: if the describe of "the second relies on destructive sampling of the soil and offline analysis of the extracted water" is suitable, and the non-destructive collection and online monitoring of the water vapour isotopic composition of soil atmosphere also can be used.

This is true, thank you! This was revised accordingly L439-442:

"*The two approaches differ in numerous aspects: while the first is non-destructive and requires on-line and continuous measurements of a few variables (i.e., water vapour mixing ratio and isotopic composition of the chamber inlet and outlet air), the second relies – with the exception of the study of Quade et al. (2019) – on destructive sampling of the soil and offline analysis of the extracted water.*"

2. Line 640-670: Péclet effect is very important for these theoretical equations, however, its definition and quantification are not explicitly named here.

Peclet effect (PE) is indeed important for quantifying the heterogeneous distribution of $\delta_L$. However, and after careful review, we did not find a study, in which PE was used to simulate $\delta_T$ and ultimately T/ET values. Dubbert et al., (2014) mentions PE as hint upon discrepancies between $\delta_L$ measurements and simulations while Wang et al. (2015) test different $\delta_L$ models against each other over the course of one day. Piayda et al. (2017) presents the PE however do not report the results, arguing that "small differences in isotopic composition were found [between the bulk leaf water and water at the evaporative sites], which were not significant for the results shown in [their] work". This is why we did not review the use of PE presently.
In order to avoid possible confusion, we removed mention to PE L150.

Dubbert, M., Piayda, A., Cuntz, M., Correia, A. C., Costa E Silva, F., Pereira, J. S., and Werner, C.: Stable oxygen isotope and flux partitioning demonstrates

understory of an oak savanna contributes up to half of ecosystem carbon and water exchange, Front Plant Sci, 5, 530, doi:10.3389/fpls.2014.00530, 2014.

Wang, P., Yamanaka, T., Li, X. Y., and Wei, Z. W.: Partitioning evapotranspiration in a temperate grassland ecosystem: Numerical modeling with isotopic tracers, Agr. Forest Meteorol., 208, 16-31, doi:10.1016/j.agrformet.2015.04.006, 2015.

Piayda, A., Dubbert, M., Siegwolf, R., Cuntz, M., and Werner, C.: Quantification of dynamic soil-vegetation feedbacks following an isotopically labelled precipitation pulse, Biogeosciences, 14, 2293-2306, doi:10.5194/bg-14-2293-2017, 2017.

3. Line 735: For the cryogenic extraction methods of xylem water, new biases might be considered, because Chen et al. (2020) found that a dynamic exchange between organically bound deuterium and liquid water during water extraction can cause the stem water cryogenic extraction error, rather than deuterium fractionation occurs during root water uptake.

Chen, Y., Helliker, B.R., Tang, X., Li, F., Zhou, Y.and Song, X. (2020) Stem water cryogenic extraction biases estimation in deuterium isotope composition of plant source water. Proceedings of the National Academy of Sciences, 202014422

We thank you for this nice new paper on issues related with extraction of water from plant tissues and have incorporated its analysis in our revised manuscript L723-729:

"*Recently, Chen et al. (2020) documented during a series of laboratory-controlled experiments that the apparent offset measured between the hydrogen isotopic composition in sap xylem and source water of different Mangrove plant species was the result of artefacts during the vacuum extraction process, rather than due to isotopic fractionation during water uptake. This could be a reason for hydrogen isotopic offsets reported elsewhere in the literature (e.g., Ellsworth and Williams, 2007; Barbeta et al., 2019). If applicable to other species, the results of Chen et al. (2020) would suggest caution in determining T/ET values based on the determination of $\delta^2 H_T$ directly from $\delta^2 H_{xyl}$ at ISS (Eq. (24b)) or considering NSS conditions using, e.g., the Craig and Gordon (1965) equation (Eq. (18')).*"